# Multisensor Systems and Arrays for Medical Applications Employing Naturally-Occurring Compounds and Materials

**DOI:** 10.3390/s20123551

**Published:** 2020-06-23

**Authors:** Rasa Pauliukaite, Edita Voitechovič

**Affiliations:** Department of Nanoengineering, Center for Physical Sciences and Technology, Savanoriu Ave. 231, LT-02300 Vilnius, Lithuania; edita.voitechovic@ftmc.lt

**Keywords:** naturally-occurring compounds, multisensor system, electrochemical sensors, blood, urea, saliva, chip, array

## Abstract

The significant improvement of quality of life achieved over the last decades has stimulated the development of new approaches in medicine to take into account the personal needs of each patient. Precision medicine, providing healthcare customization, opens new horizons in the diagnosis, treatment and prevention of numerous diseases. As a consequence, there is a growing demand for novel analytical devices and methods capable of addressing the challenges of precision medicine. For example, various types of sensors or their arrays are highly suitable for simultaneous monitoring of multiple analytes in complex biological media in order to obtain more information about the health status of a patient or to follow the treatment process. Besides, the development of sustainable sensors based on natural chemicals allows reducing their environmental impact. This review is concerned with the application of such analytical platforms in various areas of medicine: analysis of body fluids, wearable sensors, drug manufacturing and screening. The importance and role of naturally-occurring compounds in the development of electrochemical multisensor systems and arrays are discussed.

## 1. Introduction

The simultaneous detection of several analytes, or multiplex analysis, in multicomponent media is of great interest to researchers in chemistry, biotechnology, physics, molecular biology, technology, etc. It allows obtaining more information in less time about a sample under study than traditional single-target analysis. However, due to the overlapping analytical signals from several compounds, complex and often expensive instrumentation is required for this task. One example of cost-effective analytical platforms allowing the detection of several analytes at the same time, are multisensor arrays (MSAs) and multisensor systems (MSSs) that have been extensively studied and developed in recent decades. They can be applied for quality control of products (e.g., foods and beverages) at the production stage, in environmental monitoring, and, certainly for various medical purposes [1,2]. Although some sensing systems are already commercially available, the development of new devices meeting the requirements of pharmaceutical and biomedical areas is of high importance. 

In order to avoid the ambiguity caused by different terms related to MSSs, it is useful to clarify the notation used in this review. First of all, it is necessary to discuss what kind of MSS can be called as “e-tongue”. According to IUPAC, an “e-tongue” is “*an analytical instrument comprising an array of nonspecific, low-selective, chemical sensors with high stability and cross-sensitivity to different species in solution, and an appropriate method of PARC and/or multivariate calibration for data processing. Stability of sensor behavior and enhanced cross-sensitivity, which is understood as a reproducible response of a sensor to as many species as possible, are of primary importance*” [3,4]. It is further stated that: “*If properly configured and trained (calibrated), the electronic tongue is capable of recognizing the qualitative and quantitative composition of multispecies solutions of different natures*” [3]. On the other hand, when such systems are used for the purposes different from assessment of taste, the term “electronic tongue” is less convenient and “multisensor system”, “sensor array” or “smart sensors” are more relevant ones [1,5]. The same considerations can be applied for the MSSs developed for analysis of odors, known as “electronic noses” (e-noses) that can be also used for wide range of analytical tasks [6]. In current review, the term “MSS” is related to the e-tongues and e-noses functioning as arrays of non-specific sensors, and the qualitative or quantitative information about a given sample can be extracted only by multivariate data treatment. Then the term “MSA” defines an array of independent selective sensors that can separately provide the qualitative or/and quantitative information about a sample. 

To narrow the scope of this work, a brief overview of different types of sensors and their ensembles in given in Figure 1. Well-known single sensors (Figure 1a–d) are frequently used for analysis of single analytes (in some cases in a complex sample matrix) or certain physical properties; nevertheless, multiplex single sensors, suitable for detection of several analytes, also exist. Similarly, MSSs (Figure 1e–f) can be applied both for analysis of single or multiple analytes. MSAs (Figure 1g–i), in addition, allow measurements of physical properties separately or simultaneously with the detection of chemical and biological compounds. In the current review, the attention will be focused on MSSs and MSAs applied for multiplex analysis in medicine; the MSAs shown in Figure 1i will not be discussed (and will only be mentioned briefly in Section 3.2 in our discussion on wearable electronics), because the arrays of this type are closer to electronics than electrochemistry. 

A growth in the interest in the applications of various MSSs and MSAs in biomedical studies can be noted. This field has especially strict requirements for high quality of the analytical devices used in terms of biosafety and biocompatibility as well. Therefore, the priority in design and development of such analytical platforms is given to naturally-occurring compounds, since they are excellent receptors with superior selectivity for the target compounds, biodegradable and most often biocompatible in comparison with synthetic analogues; however, it should be noticed that the level of biocompatibility differs depending on the application of analytical device, and inert synthetic materials can be used in some cases. Coupling of the MSSs/MSAs with the electrochemical transducers provides many advantages comparing to other analytical methods, e.g., he capability to analyze unmodified samples, portability, and the possibility of their miniaturization. This review reports on the recent progress in the development and application of electrochemical MSSs and MSAs, based on naturally-occurring and natural compounds, in different areas of medicine (mostly, for analysis of liquid biosamples) and drug manufacturing. General principles of MSSs and MSAs functioning are summarized as well. 

## 2. Principles of MSS and MSA

### 2.1. Basics

Sensory organs, such as the tongue or nose, serve as prototypes for the design of artificial multisensor systems and arrays. For example, when one eats a food, its components are recognized by different types of taste buds on the tongue; after an interaction with organoleptic receptors, signals are sent to the brain, which processes them and sends a response to other organs so that this food can be swallowed and the corresponding enzymes be produced. Electrochemical MSSs are based on the similar approach, as shown in Figure 2. An (electro)chemical reaction occurs within each sensor/recognition element that produces a signal, which is sent to a transducer and is processed afterwards using different statistical and multivariate data treatment techniques. In general, the final result is a conclusion about the quality of the sample [1,2,5,7,8] (Figure 2a). In the case of MSA, each sensor/recognition element also senses a target compound by (electro)chemical reaction occurring within the interface of the sensor surface and the analyzed sample. Thus, the signal processing can be performed in two ways: (i) using convenient protocols for a particular method, applied for the signal registration; and (ii) using multivariate data treatments. Thereby, it is possible to obtain qualitative and quantitative information about several detected compounds as well as about the integral quality of the sample (Figure 2b).

The theoretical background of MSS development and the main signal processing methods were explicitly described in [9,10] more than 10 years ago. The authors presented several methods for processing the data sets obtained by MSSs from gaseous samples: (i) artificial neural network (ANN); (ii) flexible recognition algorithm and (iii) fuzzy logic recognition algorithm. At the same time, one of the most common methods used for processing of data obtained by the MSSs during the analysis of solutions is chaos theory [11]. Recently, many other chemometric methods have been used for the extraction of the required information about a sample, according to the particular task at hand [8,12].

### 2.2. Instrumentation

Generally, there are two main approaches to designing an MSS or MSA: (i) when several sensors are joined into one system as separate physical elements (Figure 3a) [13], or (ii) when several microelectrodes are combined into a miniaturized system of conventional electrode size (Figure 3b) [14]. 

Several conventional electrodes can be assembled into a common electrochemical cell with reference and counter electrodes, as in the commercial TS-5000Z [1] and ASTREE [15] systems, which consist of five or more working electrodes (WEs) for the analysis of liquids. Also, the panel of electrodes can be combined together with a flow system, which is typically used for the automation of the sample delivery and washing procedures [16]. Nevertheless, some MSA architectures with larger number of sensors were reported [17,18,19] (Figure 4), which can be used also in a common electrochemical cell or be incorporated into an advanced microfluidic system.

The choice of the (micro)electrode array architecture depends on a sample type and a detection method. Obviously, if the sample volume is large (i.e., given in mL), is preferable to choose conventional electrodes of normal size,, since it makes the manufacturing of an entire electrode panel simple and cost-effective; besides, the overlap of the electrochemical signals from the electrodes is also reduced. However, if sample volume is up to several µL, the only available option is a miniaturized multielectrode panel. Potentiometric MSSs and MSAs are typically used as easy-to-use and time-saving analytical platforms; however, other electrochemical methods, such as voltammetry, amperometry, or electrochemical impedance spectroscopy, can be successfully applied for various analytical tasks [1,2,3,4,5,17].

The response of potentiometric sensors can be calculated from the Nernst equation [20]:*E* = *E*_0_ + *RT*/*nF* ln*a*_i_,(1)
where *E*_0_ is the standard electrode potential, *R* is the gas constant, *n* is the electron number, *F* is the Faraday constant, and *a*_i_ is the activity of the determined ion. In the case of potentiometric sensor arrays, functioning in multicomponent media, a selectivity coefficient *k* is added to the Nernst equation [21]:*E* = *E*_0_ + *RT*/*nF* ln(*a*_a_ + k*a*_b_ + k*a*_c_ + … k*a*_n_),(2)

Voltammetric sensors demonstrate Nernstian behaviour under conditions of equilibrium between reduced and oxidized compound states [22]. The amperometric signal of the sensors is described by the Cotrell equation [20]:*I* = *nFAc*_0_(*D*/π_t_)^1/2^,(3)
where *I* is the current, *n*—the number of electrons transferred, *F*—the Faraday constant, *A*—the area of planar electrode, *c*_0_—the initial concentration of the analyte, *D*—the diffusion coefficient, and *t* is the time elapsed since the potential was applied.

Finally, the impedimetric sensor is described by an impedance measurement [23]:*Z* = *V*_0_ sin(2π*ft*)/*I*_0_ sin(2π*ft* + *φ*),(4)
where *V*_0_ and *I*_0_ are the maximum voltage and current signals, respectively, *f* is the frequency, *t*—time, *φ*—the phase shift between the voltage-time and current-time functions. The impedance is a complex value affected by multiple factors, which is described either by the modulus |*Z*| and the phase shift *φ* or, alternatively, by the real *Z*’ and the imaginary *Z*” parts of the impedance.

A great challenge in MSSs and MSAs is the development of an instrument which can control and read many channels at the same time. In the case of an e-tongue, this is easier because only 5–7 working electrodes are used in the majority of reported cases [1,2,17]. However, problems arise when it is necessary to make a microelectrode array chip with 20 or more electrodes, because the attainment of a mechanically stable microelectrode array is not such an easy task. On the other hand, signal acquisition from many electrodes is also complicated. When there are many electrodes on a chip, their electric fields overlap, interfering with each other and, therefore, more than one counter and reference electrodes are required to solve this problem. More information about the features of the design of the electrode arrays can be found in [1,2,17].

### 2.3. Materials for Electrochemical MSAs and MSSs

The general requirements for the design of an analytical system usually include: (i) as high a sensitivity as possible, (ii) cost effectiveness, (iii) portability and (iv) durability. Taking into account these requirements, researchers usually combine different types of the materials to ensure the desired properties. Thus, synthetic and natural materials are often used together in MSSs and MSAs. If a definition of the synthetic materials is obvious, it is necessary to clarify the definition of a term “natural material”. A natural product, according to The Free Dictionary [24], is “*a chemical substance produced by a living organism;—a term used commonly in reference to chemical substances found in nature that have distinctive pharmacological effects. Such a substance is considered a natural product even if it can be prepared by total synthesis*.” Therefore “natural” materials or compounds, mentioned in this review, are compounds/materials that can be produced by living organism. 

Usually, each electrode within an electrochemical MSS/MSA has the same or similar composition and architecture as conventional electrochemical sensors (Figure 5): MSS/MSA base (mandatory)Electrode material (mandatory)Layer for increasing the electroactive area (if needed)Layer for accelerating the electron transfer (if needed)Recognition element (if needed)Label (if needed)

As it can be seen, not all components are strictly required for a sensor; this depends mostly on the application and sample type, target analytes, and measurement conditions (in situ, online, in vitro, in vivo, etc.). 

#### 2.3.1. MSS/MSA Base

In order to depose or to integrate an electrode in an MSS or MSA, it has to be made of an insulating material. There are two main criteria determining the choice of the MSA/MSS base: should the MSS/MSA be rigid or should it be stretchable and/or wearable. The most widely known electrodes for solid MSAs and MSSs are made of different plastic bases in the shape of a tube, plate or strip [25]. However, recent progress in the material science has opened new possibilities in the development of stretchable and wearable sensors, which are now of great interest [26]. Along with the synthetic materials (e.g., nylon, polyester, Teflon), natural fibers play an important role in the development of wearable electronics and smart textiles. 

Cotton fiber consists of a high molecular weight polysaccharide, the linear 1,4-glucan (C_6_H_10_O_5_)*_n_*, known as cellulose, the most abundant natural polymer in the world. Cotton contains 82–96% cellulose; other components are hemicellulose (2–6.4%), lignin (0–5%) and pectin (0–7%) [27,28]. Approximately 24 million tons of cotton products are manufactured annually [27], mostly for manufacturing clothing. Cotton fibers have unique properties, such as good mechanical performance (tensile strength 300–700 MPa, specific strength 194–452 MPa, Young’s modulus 6–10 GPa, specific modulus 4–6.5 GPa, failure strain 6–8%) [29], and high absorbance rates that allow easy modification of the material with conducting compounds [30], antimicrobial reagents [31], and dyes. In addition, the fineness, durability and comfort of cotton materials make them the most popular basis for wearable e-textile development. Unfortunately, the production of the natural cotton has limited resources and low production rate, so instead of it, synthetic and regenerated cellulosic fibers can be used. The latter are obtained by chemical treatment of wood pulp, cotton linter or bamboo [32]. Such fibers, being biodegradable, have an advantage over fully synthetic fibers (as nylon, acrylic, polyester, etc.) [32]. However, the total impact on the environment caused by production of the regenerated fibers is still is dispute [33]. The most known cellulose-based regenerated fibers are viscose and cellulose acetate. Cellulose acetate is mostly applied in bioelectrochemical MSS development as a substrate for protein immobilization [34,35] or as a protective semipermeable membrane [36]. Being natural in origin, viscose provides the comfort of natural fiber, breathability, softness and durability, and can be modified both chemically and structurally in many ways.

Another example of a substrate for MSAs and MSSs is silk. Silk can be produced by a multitude of spiders and insects, including wasps, honey bees and silkworms (the most known is *Bombyx mori*, which is used for natural large-scale production of silk). Its structure and form depend on the insect species [37]. Generally, silk consists of fibrous proteins containing highly repetitive sequences of amino acids: fibroin and sericin. These proteins are arranged into fibers following a highly optimized hierarchical nanostructure, which results in the excellent mechanical properties of silk (strength 610–690 MPa and toughness 70–78 MJm^−3^) [38]. Silkworm silk is highly biocompatible and easily modifiable, therefore, it finds wide application in diverse areas of medicine, such as tissue engineering, drug delivery, and the development of (bio)sensing devices. Some reviews about the uses of silk- based materials in medicine have been recently published [37,38,39,40,41]. 

#### 2.3.2. Electrode Material

In order to reduce the costs of MSS/MSA production, non-metallic materials are often used. One of the most common materials is carbon and its allotropes (graphite, glassy carbon), composites and nanomaterials (carbon nanotubes, graphene) [1,42]. Non-conducting materials and flexible polymers, covered with conducting films are also used as sensor substrates for producing the miniaturized electrode arrays [1,9,43,44,45,46,47]. Natural fibers (cotton, silk, cellulose (Section 2.3.1)) can be modified with conducting polymers to form composites, used as electrode materials, with exceptional electric properties [48]. Such materials can find wide use in biomedicine, because they can be applied separately as wearable sensors or be integrated into smart textiles.

#### 2.3.3. Layer for Increasing the Electroactive Area and/or Signal Amplification Element

This layer is required when the electroactive area of an electrode material is smaller than the geometric one. The main purpose of this layer is the signal amplification, which can be done either by increasing the electroactive area or by decreasing the overpotential. Thus, it is often applied when either non-metallic or metallic substrates, modified with specific materials (e.g., gold, modified with sulphur containing non-conducting compounds or proteins) are used. In most of cases, some conducting nanomaterials are used to increase the electroactive area, such as metal (typically noble) nanoparticles [1,43,49,50,51,52], carbon nanomaterials (nanoparticles of boron-doped diamond, carbon nanotubes, graphene) [43,53,54,55], other non-metallic nanoparticles [56]. Usually, the use of carbon nanotubes and graphene in e-noses and MSA is reported [54,55].

To immobilize nanomaterials onto the electrode surface, some additional compounds are usually required. Most of the natural polymers, e.g., gelatin, collagen, chitosan, cellulose and its acetate are applied for this purpose [57]. Chitosan, a biological polysaccharide polymer, is widely used in MSAs and MSSs [58,59,60], while other naturally-occurring compounds are used mainly in studies of single analyte sensors [57].

#### 2.3.4. Layer for Accelerating the Electron Transfer

This layer is also related both to the signal amplification and to the catalysis of an electrochemical process. The compounds accelerating the electron transfer are often called electrochemical mediators [61]. Generally, they reduce the activation energy and, as a consequence, decrease the overpotential of oxidation or reduction of the target compounds. However, mediation mechanisms can be different, e.g., (i) elimination of the kinetic inhibition, associated with the electron transfer at the electrode/electrolyte interface; (ii) a mediated electron transfer, that occurs against a potential gradient, thus, lower potentials are needed [62]. Both organic and inorganic compounds can serve as mediators. Although redox mediation is well known, in recent years, several new types such mediators were designed and studied [62,63]. As stated above, mediators are also used in MSAs and MSSs [43,59,64].

This layer can be also composed of naturally-occurring compounds. Our recent works reports an application of several compounds from vitamin B group—riboflavin (B_2_) and folic acid (B_9_)—for biosensing [57,65,66,67,68]. These vitamins should be polymerized because they are soluble in water and will leak from the electrode surface in their monomeric form [65,67]. The advantage of the accelerating layer is that it can immobilize the layer for increasing electroactive area and work as a redox mediator. Moreover, it is important to highlight that such polymers can be used for in vivo measurements because they decompose into monomers that are safe for living organisms. However, to the best of our knowledge, these vitamins have not been used in MSSs and MSAs yet.

#### 2.3.5. Recognition Element

If the MSAs or MSSs are used for evaluating some chemical properties of a sample and not the physical ones (temperature, strain, movement and etc.), a recognition element becomes one of the most challenging components of the system. The recognition element enters into a specific reaction with a target molecule, emitting or absorbing the electrons, or in some cases changing the electrochemical parameters during the measurements (e.g., for affinity (bio)sensors). Most recognition elements are biological molecules, or bioreceptors; therefore they are natural compounds. The largest group of them is proteins: peptides, olfactory receptors, cell receptors, enzymes and immunoglobulins. Other bioreceptors such as the nucleic acids (oligonucleotides, aptamers), carbohydrates (different lectins [69], concavalin A is the most known one), organic molecules such as natural ionophores [70] (valinomycin for K^+^ ions, and nonactine for NH_4_^+^ ions) can be used for the recognition of a target compound. Along with the natural compounds or polymers, phages [71], bacteria, whole cells [72], even tissues [73] and organs [74] are used for sensing by MSS/MSA. Also, molecularly imprinted polymers (MIP), being of non-biological origin, can be used as recognition elements [75,76]. Advantages and specificities of each mentioned group of recognition elements is described in detail in the review by Justino et al. [71]. Enzymes and antibodies are the most frequently used bioreceptors in MSS and MSA, as well as in single-analyte biosensors; their choice depends on a target analyte [77]. Morales and Halpern developed a flowchart to simplify the choice of an appropriate biorecognition element (Figure 6).

#### 2.3.6. Label

Biosamples, especially ones selected for diagnostic needs, typically contain an extremely low concentration of the target analyte, i.e., in nano-, pico-, even in femtomolar concentration range. Thereby, such a negligible amount of the analyte cannot induce significant electrical changes, which can be registered directly by recognition element-target compound interactions. In this case, an additional signal amplification element, called a label, is integrated into MSA or MSS during the analysis. Usually, the signal amplification is needed for the affinity biosensors. There are several types of labels that can significantly amplify the electrochemical response [78]. The first and, possibly, the most common, is an enzymatic label (mostly known as “enzymatic tag” or “enzyme conjugate”): a redox enzyme characterized by the high turnover number (e.g., horse radish peroxidase (HRP), glucose oxidase (GOx), tyrosinase). The enzymes of this kind can produce a high concentration of electroactive compounds in a short time period; then, these compounds undergo oxidation or reduction within the electrode material. The second type of labels are different nanoparticles and nanomaterials characterized by high electroconductivity, e.g., gold or silver nanoparticles [78]. 

## 3. Application of MSSs and MSAs

Most of the MSSs/MSAs discussed here have been used for medical diagnostics; typical targets of analysis were body fluids, such as blood or its serum, saliva, urine, tears, cerebrospinal fluid (CSF), gastrointestinal fluid (GIF), wound exudate (WEx) and sweat. These analytical platforms can also be successfully applied in pharmaceutics. The latest achievements of MSSs/MSAs applications in pharmaceutical applications for drug manufacturing and screening are summarized at the end of this Section.

### 3.1. Analysis of the Body Fluids

#### 3.1.1. Blood

Among the various fluids produced by human body, blood is the most frequently used one for diagnostic purposes, because it contains different metabolites, nutrients, proteins, which are transported to and from all organs and tissues. As a consequence, blood contains many biomarkers of various diseases, and should be analyzed following good clinical practice principles [79]. At present, the glucose, ionic content, numerous cancer biomarkers, are, possibly, the most attractive targets of blood analysis performed by MSSs/MSAs. Two types of the blood tests are generally used: (i) direct, when a blood drop or a sample of the venous blood is analyzed using (bio)sensors, (ii) indirect, when the concentration of target compounds in the blood is assessed by measuring some physical parameters, e.g., the intensity of the blood flow [80], dielectric properties [81] or skin temperature [82]. Despite the fact that the direct blood analysis is invasive and potentially raises certain health risks, it still remains the most accurate diagnostic procedure. Revised MSSs/MSAs used for blood analysis are presented in Table 1.

Affinity biosensors are the most common multisensor platforms applied for the screening of cancer biomarkers. Electrochemical enzyme-linked immunosorbent assay (ELISA), based on the amperometric registration system, has become an advantageous biotechnological tool in scientific research and clinical diagnosis [83,84] (Figure 7). Its high sensitivity and specificity towards target proteins—antigens, in the case of conventional optical ELISA (which can quantify biomarkers of diseases in the pg/mL range), and additional features of electrochemical assays (e.g., lower sample volume, low-cost instrumentation, short time of analysis) contributed to the commercial success [83]. Also, the electrochemical ELISA is suitable for the analysis of known cancer biomarkers, and, according to a particular task, can determine up to 96 targets simultaneously [83]. 

Similar amperometric registration systems were applied in other immunosensor arrays. The design of the microarrays and the signal amplification are on the main focus of the development of cost-effective and accurate point-of-care devices [85,86]. Some devices, such as an amperometric immunosensor based on a screen-printed (SP) chip for simultaneous quantification of carbohydrate antigen 19-9 (CA19-9), and cancer antigen 125 (CA125) [34], a photoresist-patterned microfluidic paper-based analytical device with amperometric detection system for α-fetoprotein (AFP), carcinoembryonic antigen (CEA), CA125, and cancer antigen (CA153) [87], successfully demonstrated their potential applicability in clinical practice to identify cancer biomarkers in serum samples taken from cancer patients. Differential pulse voltammetry (DPV) was also used for registration of CEA and mucin-1 (MUC1) in blood serum by dual-target electrochemical aptasensor [88]. Similar analytical approach was applied for simultaneous detection of CEA and neuron-specific enolase (NSE) with multi-parameter paper-based electrochemical aptasensor [89].

Another type of promising affinity biosensors are impedimetric biosensors, which are based on the registration of resistance changes determined by the binding of target molecules to the modified electrode surface [84,90]. Simultaneous detection of three ovarian cancer biomarkers—CA125, CEA and human epididymis protein 4 (HE4))—was performed by a low-cost impedimetric biosensor based on silicon chips with nanoscale-gapped interdigitated electrode arrays modified with the monoclonal antibodies. This biosensor allowed multiplexed detection of cancer biomarkers down to 0.1 ng/mL [91]. Prostate-specific antigen (PSA) and NSE were quantified by an immunosensor based on antibody modified gold electrodes, using methods of electrochemical impedance spectroscopy (EIS) and impedance time (Z-t) measurements [92]. Another impedimetric MSA measurement strategy is the registration of the resistance changes caused by the enzymatic activity of a target protein. For example, multiplex quantification of ovarian and colorectal cancer biomarkers: matrix metalloproteinases 2 and 7 (MMP-2 and MMP-7, respectively) was performed by an electrode array modified with peptide-specific substrates for MMP-2 and MMP-7. Depending on the activity of the proteases, the peptides underwent proteolytic hydrolysis causing a decrease of the resistance [93]. The bio-e-nose consisted of 30 types of the olfactory receptors from HEK-293 cell line, which were immobilized on single-walled carbon nanotube (SWCNT) field effect transistor (FET). The system was used for the perception of heptanal as a lung cancer biomarker in human blood [94]. Along with the analysis of cancer biomarkers, electrochemical MSAs can be applied for blood screening for the biomarkers of hypertension [95], cardiovascular disease [96,97], for evaluation of the concentration of purine nucleotides [98], metal ions [99], urea, creatinine and uric acid [100], etc.

Recently, a novel platform named “plug-n-play digital microfluidics” was proposed for the multiplex quantification of glucose, β-ketone and lactate in blood samples [16]. The platform has specific features such as a high reconfigurability and automated fluidic operations using generic device architecture. This construction of the device allowed combining several electrochemical (bio)sensors and operating them depending on needs. An interesting area for the development of MSAs based on electrochemical detection system is monitoring of the banned compounds in blood samples of the athletes according the anti-doping rules [101]. However, to the best of our knowledge, no MSAs or e-tongues have been suggested for this application. 

Meanwhile, non-invasive monitoring of glucose dynamics in blood remains a “hot” topic of 21st century, because the diabetes are one of the most frequent metabolic disorders, which are associated with improper regulation of blood glucose concentration [102]. Natural compounds serve as a base for the electrode arrays, as in the case of the wearable electronics (see Section 2.3.1 and Section 3.2). MSSs, based on metallic electrodes that record the dielectric properties of skin, can evaluate blood glucose level with acceptable accuracy, therefore, they have great commercial potential [103,104,105,106,107]. 

#### 3.1.2. Urine

Urine is a body fluid generated by the kidneys. The urine of healthy individuals is transparent and sterile. However, depending on the disease, urine can become cloudy, or have the atypical color and odor, that are related to changes in the urine content: some metabolites (e.g., urea, uric acid, creatinine, bilirubin, glucose), numerous proteins, including N-acetyl-β-D-glucosaminidase, IL18, albumin, neutrophil gelatinase-associated lipocalin, urinary liver-type fatty acid binding protein, cystatin, and ionic composition as well [86,109]. Thereby, the diagnosis of kidney dysfunctions and injuries, arthritis, diabetes, cardiovascular diseases can be carried out by analysis of urine samples. Probably, the most convenient multiplex method for the assessment of cancer biomarkers (or other proteins) in urine, as well as in blood, is the electrochemical ELISA. However, there is an on-going effort to improve the registration and sample delivery systems of ELISA platforms [86]. Recently, a novel fluidic-based DPV ELISA platform for estimating the protein markers of bladder cancer (nuclear mitotic apparatus protein 1 (NUMA1) and complement factor H-related 1 (CFHR1)) was proposed [110]. The new platform exhibited linearity over the 1–100 ng/mL range of target proteins, and the limits of detection (LODs) in the diluted urine samples were 1.29 ng/mL and 0.97 ng/mL for NUMA1 and CFHR1, respectively. Simultaneous multiplex detection of urinary pathogens (bacterial nucleic acid (16S rRNA)) and lactoferrin by an amperometric immunosensor was demonstrated as a more promising and informative tool for diagnosis of the urinary tract infection than the current standard method [111]. Impedimetric quantification of C-reactive protein and interleukin 6 (IL6) by a novel portable dipstick-type immunosensor, based on molybdenum electrodes on nanoporous polyamide substrate, was reported in [112]; both proteins were detected in human urine samples with a LOD of 1 pg/mL. The ionic composition of urine can be useful for evaluating a patient’s general condition and diagnosing certain metabolic diseases [113]. Recently, cost-effective potentiometric e-tongues were applied for evaluation of ionic composition of patients’ urine samples; their potential applicability for express diagnosis of urolithiasis [114], and indirect assessment of the prostate cancer [115] was demonstrated. A combination of an e-nose and a potentiometric e-tongue, composed of the metalloporphyrin-based sensor array, was successfully used for evaluation of pH, specific gravity, and presence of blood in urine samples, collected from children, affected by kidney diseases, and a healthy control group [116]. Integration of biosensors in the electrode arrays usually improves the performance of the e-tongue. For example, a potentiometric e-tongue, consisted of urease-modified ammonium along with hydrogen and other ion-selective electrodes (ISEs), was applied for the assessment of the urea concentration in urine samples [117]. Such a bio-e-tongue allowed not only quantifying urea in the complex media, but also evaluating the content of interfering ions simultaneously. The bio-e-tongues with similar combination of ISEs and additional potentiometric biosensors with urease and creatinine deiminase were successfully tested for the simultaneous determination of urea, creatinine, and ions in urine and dialysate fluid samples [118,119].

Another cost-effective platform for the multiplex assessment of small organic metabolites is a paper-based microfluidic device [120]. A paper-based microfluidic device consisting of eight independent bioamperometric sensing modules was used for the simultaneous quantification of glucose, lactate and uric acid in urine samples [121]. Each module included a paper channel, patterned with hydrophobic wax, and three screen-printed carbon electrodes (SPCEs) on the test zone of the paper channel. Each working electrode was modified with the corresponding redox-enzyme. This approach allowed the quantification of glucose, lactate and uric acid in concentrations up to 20, 25 and 10 mM, respectively; the analysis took two minutes. Glucose, creatinine, and uric acid levels were simultaneously measured in real urine samples by a disposable bioamperometric paper-based analytical device [122], which also demonstrated a great potential for various medical applications. Urine analysis can be useful not only for the diagnostic or wellbeing needs; urine screening for narcotic drugs and their metabolites, performed by electrochemical systems, was also reported in the literature. For the first time, a voltammetric MSA based on an array of SPCEs modified with gold nanoparticles and antibodies was used for multiplex quantification of morphine, tetrahydrocannabinol, and benzoylecgonine in spiked urine samples [123]. Reviewed MSSs/MSAs used for urine analysis are presented in Table 2.

#### 3.1.3. Saliva

Saliva samples have attracted high interest in medical diagnostics and health monitoring because of the simplicity of their collection and the variability of the different biomarkers they contain. Saliva consists of about 2000 proteins, so which some 26% can be also found in blood. This variability of the biomarkers makes the use of saliva for point-of-care applications promising [124]. Typically, uric acid, glucose, lactate dehydrogenase, α-amylase, albumin (and some other proteins as lipocalin, cystatin C, etc.), cortisol, testosterone and pH were quantified for the diagnosis of metabolic disorders, such as diabetes, but also some types of cancer, stress and decalcification [109]. Salivary biomarkers are routinely measured using common techniques, such as radioimmunoassay (RIA), MS, DNA-based techniques, optical and electrochemical ELISA [124,125]. Furthermore, some promising electrochemical affinity biosensors were suggested for the multiplex assessment of saliva [124]. Two oral cancer biomarkers (interleukin 8 (IL-8) mRNA and IL-8 protein) were simultaneously quantified by the method of chronoamperometry. For this task, the array of 16 gold electrodes was modified with co-polymerized polypyrrole streptavidin-dendrimer nanoparticles and bioreceptors: hairpin probe, monoclonal antibody for mRNA, and IL-8. The obtained MSA, used in the multiplexing mode, has shown an LOD of 3.9 fM for IL-8 mRNA, and 7.4 pg/mL for IL-8 protein in patients’ saliva samples [126]. The same oral cancer biomarkers were also quantified by the amperometric magnetic biosensor consisting of two SPCEs (the carbon counter electrode, and the Ag pseudo-reference electrode), and magnetic particles modified with hairpin DNA probe for mRNA and monoclonal antibody for IL-8 protein [127]. A new MSA provided high sensitivity and selectivity towards the target analytes. The LOD of 0.21 nM for IL-8 mRNA and of 72.4 pg/mL for IL-8 protein were achieved in undiluted saliva samples. A MSA, capable of detecting mutations in the epidermal growth factor receptor (EGFR) gene directly in saliva samples, was proposed to identify epithelial cancers, particularly non-small cell lung carcinoma [128]. Simultaneous detection of the L858R point mutation in Exon21 and Exon19 deletion was performed by the MSA in two steps: (i) the release of the target biomarkers from the saliva samples by induced electric field, and (ii) the specific sandwich hybridization assays performed onto a conducting polymer. The EGFR mutations were measured by chronoamperometry, when the labeled with the redox-enzyme detector DNA probe was bound to the DNA of the cancer cells [128]. 

Dual SPCEs modified with 4-carboxyphenyl-functionalized double-walled CNTs were used to fabricate an amperometric immunosensor for simultaneous detection of interleukin 1β (IL1β) and tumor necrosis factor α (TNF-α) in human saliva samples [108]. The relevant monoclonal antibodies for capturing the target analytes were immobilized onto working electrodes modified with the commercial polymeric coating Mix&Go™; HRP-streptavidin conjugates were used to amplify the signal. The proposed amperometric immunosensor was able to detect IL-1β and TNF-α in saliva samples with the LOD of 0.38 pg/mL and 0.85 pg/mL, respectively. 

The quantification of hormones present in saliva can be also interesting from a diagnosis point of view. Multiplex determination of human growth (hGH) and prolactin (PRL) hormones was performed by DPV using CNT–SPCEs modified with poly(ethylene-dioxythiophene) (PEDOT), gold nanoparticles, and relevant monoclonal antibody. The antigen-antibody affinity reactions were monitored by measuring the decrease in the DPV oxidation response of the redox probe—dopamine. The LODs for hGH and PRL were determined as 4.4 and 0.22 pg/mL, respectively [129]. A similar approach was applied for the simultaneous determination of two hormones: ghrelin (GHRL) and peptide YY (PYY). that play important roles in the regulation of hunger and satiety feelings. Quantification of these hormones was performed by the dual SPCEs modified with reduced graphene oxide and the relevant monoclonal antibody [130]. For this purpose, competitive immunoassays were used; the antibody-antigen interactions were monitored by DPV upon addition of 1-naphthyl phosphate. The proposed method demonstrated its applicability in hormone quantification, presented in concentrations up to 100 ng/mL in saliva samples. 

Furthermore, an important object for analysis is the ionic composition of saliva. The secretion of saliva is regulated by both sympathetic and parasympathetic innervation; the dysfunction of these two systems can be triggered by certain diseases [131]. The evaluation of the composition of saliva in terms of the selected ions: Na^+^, K^+^, Ca^2+^, Mg^2+^, Cl^−^ and HCO_3_^−^, could help in the monitoring of the mentioned dysfunction. Thus, a simultaneous determination of these ions in saliva samples using potentiometric MSA, consisted of ISEs (partially with natural ionophores), was proposed [131,132]. The results, obtained by the MSA, were in good agreement with the data obtained by the reference methods and have shown potential applicability of this potentiometric MSA for direct monitoring of the changes of electrolyte concentrations for athletes, according to the intensity of their physical activity. Reviewed MSSs/MSAs used for saliva analysis are presented in Table 3.

#### 3.1.4. Tears

Tears are secreted by the lacrimal glands to lubricate and cleanse the surface of the eyeballs and inner surface of the eyelids. Their functions are not limited by the lubrication of the eyes, or removing irritants; tears also contribute in the local immunity. Therefore, the qualitative and quantitative analysis of lactoferrin, lysozyme, secretory immunoglobulin A, lipocalin-1, lipophilin, lacritin, proline-rich proteins, MMP-9, Th1 and Th17 cytokines in tear samples can help to diagnose many ocular and systemic diseases including diabetes, cancers, and allergies [109]. Most of these biomarkers are analyzed by common “omics” techniques [133]. The electrochemical affinity biosensors can be suitable for multiple protein biomarker assessment in tears; however, none of them were suggested for this kind of analysis yet. The well-known analyte present in tears, is glucose, because its concentration correlates with the one in blood. Therefore, its level can be evaluated from tears, avoiding the application of invasive methods [134]. The analysis of tears in vitro is quite complicated and is associated with the difficulty to collect the required amount of the sample and evaporation of tears during the collection of the sample; overall, the collection methods are quite challenging in this case. Therefore, the concentration and composition of a sample can be significantly affected [134]. To avoid this issue, most of the reported (bio)sensors for the analysis of tears are wearable, made in the shape of a contact lens [133,134]. This analytical platform contains all necessary (bio)sensing components, including the data processing and power sources, that leads to a complex design and fabrication process (Figure 8). 

Different optical methods were used earlier to register the signal from the sensitive part of above-mentioned contact lenses [134,135,136]. However, electrochemical methods have also demonstrated the necessary sensitivity, linearity and accuracy levels, appropriate for this task. Within such analytical platforms, natural compounds can serve as flexible supports (e.g., natural hydrogels) for the deposition of the sensing elements, or can be used as bioreceptors. One of the pioneering wearable MSAs for glucose quantification in tears was an amperometric MSA placed on the commercial contact lens. It consisted of two working metallic electrodes: one was modified by GOx, the other was used as a bare electrode for the assessment of the reference response [136]. The proposed design of the MSA allowed fast measurements of the glucose concentration and eliminated the impact of interfering chemicals, found in tears, such as ascorbic acid, lactate and urea. Similar biorecognition system was applied in the development of a “smart” electrochemical MSA within the soft contact lens. The MSA was composed of the transparent and flexible electrodes based on graphene-AgNW hybrid and it was suggested for the measurements of glucose and intraocular pressure in vivo and in vitro, although their simultaneous measurement is not possible yet [137]. 

#### 3.1.5. Cerebrospinal Fluid

Cerebrospinal fluid (CSF) is a body fluid found in the brain and spinal cord and produced from arterial blood by the choroid plexuses of the lateral and fourth ventricles of the brain. CSF is typically used for the diagnosis and monitoring of chronic neurological disease [109]. The main target components in the CSF samples are proteins. The quantification of total concentration of protein Tau (tTau) is widely used, because the fluctuations of its level are associated with the neurodegenerative diseases [138]. Furthermore, the quantification of cystatin C, β2-microglobulin, nerve growth factor precursor, transthyretin, retinol-binding protein, ApoA1, ApoE, granin-like neuroendocrine precursor, pigment epithelium-derived factor, retinol-binding protein, haptoglobin and β -amyloid 1–42 can be used for the diagnosis of Alzheimer’s disease and amyotrophic lateral sclerosis [109,138]. Monoamine neurotransmitters play the crucial role in functioning of cardiovascular, renal, hormonal systems, and they are related to the numerous psychotic and neurodegenerative diseases [139]. Therefore, qualitative and quantitative analysis of the above-mentioned essential organic compounds in CSF is of great clinical significance despite their low concentrations. 

The multi-walled carbon nanotubes (MWNTs)-ZnO/chitosan composites modified SPEs were applied for the simultaneous determination of norepinephrine and serotonin using square wave voltammetry (SWV) [140]. The peak potentials of norepinephrine and serotonin were separated and detected at ca. 90 mV and 280 mV, respectively. Similar approach was applied for the development of MWNTs-SiO_2_-chitosan modified SPEs for simultaneous determination of dopamine and serotonin [141]. The proposed electrochemical sensors were both tested for simultaneous detection of neurotransmitters at micromolar levels in rat CSF samples; the successful results of these studies open new perspectives for this approach in medical diagnosis.

#### 3.1.6. Gastrointestinal Fluid

Gastrointestinal fluid (GIF) is secreted by gastrointestinal epithelial cells. GIF contains proteins, hormones, ions, small organic compounds that ensure the digestion of food. Several gastrointestinal disorders, such as irritable bowel syndrome and chronic constipation, are frequently diagnosed; therefore the monitoring of gastrointestinal environment (including GIF, the shape of gastrointestinal cavity and its possible deformations, the motor activity of gastrointestinal tract) could play a vital role in the diagnosis and treatment of certain disorders [142]. 

The most convenient platforms for gastrointestinal environment analysis are ingestible electronics [143,144,145]. Various biomarkers (e.g., some proteins, DNA, RNA, gases), pH, microbiota, food content, as well as physical parameters (temperature, electrophysiology, pressure, structure, and motion) can be assessed by such devices [143]. Ingestible electronics can be in either encapsulated or tableted form. Generally, they are composed of a power supply, signal transduction elements, sensing and controlling units [143]. Depending on the task, such devices can be featured by the high resolution cameras, light-emitting diodes or controlled drug releasing units [143,146]. Electrochemical, electromagnetic, optical or acoustic signal registration methods are implemented in the sensing units [143]. In contrast with most analytical devices, ingestible electronics should work under difficult conditions: at high pH values, in the presence of digestive enzymes, bile, and mucus. Therefore all components of these devices should pass numerous requirements for being safe to use: be inert towards degradation or generate degradation products safe for organisms, and also they have to be sensitive and selective enough for the sensing task [147,148]. 

Almost all ingestible electronics are multisensing platforms that provide measurements of physical and chemical parameters. The natural materials can be used as flexible substrates (gellan gum, alginate, chitin, silk fibroin, pectin), conductors (reflectin, alginate, chitin), dielectric materials (albumin, sucrose, gelatin), thin-film transistors (indigo, β-carotene), components for the power supplies or fuel cells (melanin, GOx) [149,150]. An MSA based on thermal conductivity sensors for measurements of CO_2_, H_2_ and O_2_ in gastrointestinal tract was suggested in [147]. It was incorporated into an ingestible capsule together with the wireless signal transduction unit and power supply; the measurements were performed in vivo on a healthy volunteer. The obtained results proved that the developed ingestible device was accurate and safe for monitoring the diet impact on individuals or for applying it as diagnostic tool for the gut. The electrochemical e-tongue for non-specific characterization of gut fluids was incorporated into an ingestible capsule; the results were promising as well as in the previously mentioned research [151].

An interesting approach for the GIF analysis was suggested in [145]; the MSS was based on fully edible electrochemical sensors, integrated directly in food. The electrodes for such sensors were made of natural edible components: the activated charcoal used as the conductor and mixed with plant oils (e.g., olive and corn oils) served as pasting liquids. The electrodes were placed into hollow-shaped foods (e.g., vegetables and penne noodles), suitable as edible insulating tubes. Multiplex measurements of ascorbic acid, dopamine and acetaminophen by means of chronoamperometry, SWV and DPV were demonstrated in the spiked artificial GIF. When these electrodes were coupled together with different enzyme-rich plant tissues, they obtained the biocatalytic properties and functioned as amperometric biosensors. For example, the electrode coupled with the tyrosinase-containing tissue of mushroom, was applied for the quantification of phenolic compounds; if a horseradish tissue was used instead of the mushroom, the modified electrode was able to detect hydrogen peroxide [145]. Although the characterization and the tests of this analytical platform were performed by the external electrochemical station, the obtained results are important for developing of edible and ingestible devices for diverse biomedical applications. 

#### 3.1.7. Stools (Faeces)

Human stools (faeces) are not an actual body fluid (at least, due to their physical state), but it is an important object for analysis to diagnose of a colorectal cancer and parasite invasions [109]. In the case of colorectal cancer, the analysis of extracted DNA fragments, fibrillin-1, tumor M2 pyruvate kinase and also some volatile organic (VOCs) compounds can help to recognize tumors [109]. Some very recent findings about gut microbiota have also attracted the interest to the analysis of the stool samples [152]. The contribution of specific bacteria in the progression of certain systemic inflammatory diseases, such as inflammatory bowel disease, multiple sclerosis, systemic inflammatory arthritis, asthma, and non-alcoholic fatty liver disease, has been reported recently [153]. Thus, the 16 small ribosomal subunit RNA (16S rRNA) was also included in the list of target analytes, potentially indicating the bacteria content in gut, and determining the pathogenic ones [153]. 

The analysis of stools is performed in vitro by several well-established clinical tests [154]. Nevertheless, some novel analytical platforms for analysis of stool were proposed, demonstrating satisfying results from the point of view of further medical application and commercialization. An amperometric MSA, based on an immunosensor, was proposed for detection of *Salmonella typhimurium* and its DNA, and was tested on the human stool samples [155]. The MSA provided the LOD of 1 cfu/mL for bacteria detection and sensitivity to DNA in the range of 0.002–200 μM. The recent development of the analytical platforms on the base of artificial or biological pores with nanoscale dimensions was proved to be a successful strategy for the detection of molecules the compatible size, as DNA, RNA, peptides, proteins and polysaccharides [156]. The protein aerolysin, produced by bacteria as a β-pore-forming toxin, was applied in the electrochemical detection of botulinum toxin type B by discriminating enzymatically cleaved peptides from a synaptic protein synaptobrevin 2 derivative [157]. The proposed analytical platform allowed express (up to several minutes) multisensing of the peptides by registering the currents, produced by the targets at subnanomolar levels, by the patch-clamp amplifier. Unfortunately, this method was not tested on real stools or serum samples, but it can be potentially applied in medical diagnosis for sensitive detection of neurotoxins. A very attractive idea is to diagnose cancer without using the invasive methods of sample collection, e.g., biopsy [158]. Thus, the analysis of VOCs emitted by stool can complement the common methods for diagnosis of the gastrointestinal tract disorders. The applicability of electrochemical e-noses for the diagnosis of colorectal cancer [159,160], inflammatory bowel disease, irritable bowel syndrome, infectious diarrhea and celiac diseases [161] was already demonstrated; however, no bio-e-noses have been applied for the assessment of VOCs emitted by stools yet. 

#### 3.1.8. Wound Exudate

Wound exudate (WEx) has begun to be considered as a promising object in diagnosis, as more information about its protein content has been discovered [162]. Analysis of the WEx can be used for the diagnosis of the non-healing wounds and monitoring of the healing process of a wound in overall [163]. At present, there is no single biomarker for the diagnosis of non-healing wounds; however, several proteins—zyxin and IQGA1—have been suggested for this purpose as the most promising biomarkers for impaired cutaneous wound monitoring [162]. Also, angiopoietin-2, epidermal growth factor, TNF-α, transforming growth factor, vascular endothelial growth factor [164], C reactive protein, proteolytic enzymes (MMP-9, MMP-2, elastase), and other proteins, bacteria [165], temperature, uric acid, lactate, and pH [166] are the targets to be assessed for different kinds of wounds. There are two possible ways of monitoring the wound healing progress: (i) analysis of the exudate, collected from the wound; and (ii) incorporation of an analytical platform into the bandage, which covers the wound and analyses exudate in situ. The first way is more informative, since sensors are available for the quantification of the proteins and small molecules [163,167]. However, for certain types of wound it is nearly impossible to collect the sufficient amount of the WEx (e.g., in the case of dry diabetic ulcers), and the method of the exudate collection impacts the results of the analysis [168]. Therefore, the second option is more preferable, since it provides the real-time evaluation of the wound healing progress by monitoring of pH, uric acid, lactate, temperature, and/or moisture [169]; nevertheless, it is less informative at the same time.

Electrochemical MSSs have not been widely applied in the analysis of the wound fluids yet. Typically, HPLC, 2D-gel electrophoresis, mass spectrometry (MS) and multiplex optical ELISA are used for the multiplex assessment of the proteins in the WEx [170]. Though, several MSAs based on immunoanalytical platforms were suggested for wound protein quantification. Triggering receptor-1 expressed on myeloid cells (TREM-1), MMP-9 and N-3-oxo-dodecanoyl-l-homoserine lactone (HSL), as three known infection biomarkers in wound fluid, were quantified by impedimetric MSA based on gold SPEs modified with a relevant antibody [171]. The analysis was performed in less than an hour, achieving LOD of 3.3 pM, 1.1 nM and 1.4 nM for TREM-1, MMP-9 and HSL, respectively; no time-consuming and sophisticated sample preparation was needed. PS@PDA-metal nanocomposites based on graphene nanoribbon-modified heated SPCEs were used for the fabrication of an ultrasensitive electrochemical immunoassay for rapid detection of IL-6 and MMP-9 by SWV [172]. The LODs were achieved to be 5 fg/mL and 0.1 pg/mL, respectively, and analysis time was 2 min [172]. Unfortunately, this proposed MSA was not demonstrated in the analysis of real samples, despite the fact it has shown very promising results. Other electrochemical biosensors, based on enzymes, antibodies and aptamers were suggested for a single analyte quantification in the WEx. A few recent reviews on such kinds of (bio)sensors can be found in the literature [166,173,174,175]. 

Some MSAs, based on flexible wearable electronics, for in situ measurement of wound pH, uric acid, temperature, and/or moisture were reported [169,176,177,178,179,180,181]; unfortunately, no natural compounds were suggested as bioreceptors in such MSAs. In several studies, discussed below, natural compounds were used as the base of a MSA. For example, a MSA based on commercial palette paper and incorporated in a wound dressing, was proposed in [182]. The MSA, consisting of a flexible array of pH-sensitive electrodes, provided measurements of pH at multiple sites of one wound. The SPCEs coated with a conductive proton-selective polyaniline (PANI) membrane were used as WEs. This low-cost potentiometric MSA was able to measure pH in the range of 4–10 pH, with an average sensitivity of −50 mV/pH. The biocompatibility of the MSA was confirmed by tests with human keratinocyte cells, and can be potentially applied for in vivo measurements. A similar MSA was built on a cotton base. PANI and PEDOT: PSS composites along with Ag/AgCl quasi-reference electrode were applied as solid state pH potentiometric sensors on the wound dressing [183]; this approach showed its effectiveness in monitoring of pH changes in different sites of the wound.

Following the progress in analytical chemistry, Dargaville et al. presented their view of the future evolution of MSA/MSS, applicable for wounds [181] (Figure 9). Ideally, the future MSA/MSS would be like smart bandage, which is able to monitor pH changes, bacterial grow, non-healing and healing biomarkers. 

#### 3.1.9. Other Body Fluids

Other body fluids such as seminal fluid, nipple aspirate fluid, interstitial fluid, nasal lavage fluid, bronchoalveolar lavage fluid, etc., are less extensively studied by researchers in terms of the development of a specific method for their analysis. However, these fluids can be also used for the diagnosis of some specific diseases. 

Seminal fluid is a complex mixture of lipids, sugars, small metabolites, proteins and ions; however, for now, only Dkk-3 protein quantification in this body fluid is performed for prostate cancer diagnosis [109]. The additional informative properties to be assessed include viscosity, volume, sperm concentration, etc. [184]. 

Nipple aspirate fluid contains information about hormone and protein content, which could be suitable for the evaluation of womens’ health. Typically, the biomarkers found in nipple aspirate fluid are some micronutrients (tocopherols, cholesterols, carotenes), hormones (estradiol, estrone, progesterone, testosterone), carbohydrate antigens, urokinase-dependent plasminogen activator and plasminogen activator inhibitor [109,185]. All of them can be used for detection of the breast cancer at any stage and for women from different age groups [109]. 

Interstitial or extracellular fluid is a solution that surrounds cells and extracellular matrix of a tissue [109,186]. This fluid consists of the ions, proteins, nutrients, which diffuse the within extracellular matrix as metabolic products; the evaluation of their content might be useful in diagnosing and monitoring of some diseases. Most of studies on interstitial fluid are related to the diagnosis of cancer, because proteins secreted by tumor cells have significantly higher concentrations in tumor intestinal fluid compared to those in blood [109]. 

Nasal secretions provide valuable information on nasal pathophysiology. However, according to Riechelmann et al. [187], the published data on biomarkers’ concentrations in nasal mucus liquid are remarkably inconsistent, and this bias is caused by different sampling techniques, that have not been systematically evaluated yet. Nevertheless, several proteins were detected: α2-macroglobulin, lactoferrin, lactate dehydrogenase, IL1β, IL8, TNF-α, eosinophil cationic protein, and tryptase [187], cystatin SN, β2-microglobulin and toxic compounds from the environment [188]. Following the burning topic of COVID-19, nasal lavage fluids as well as bronchoalveolar lavage fluid were used to detect SARS-CoV-2 virus. Wang et al. showed that the analysis of nasal secretion samples is not that precise as bronchoalveolar lavage fluid, but still quite useful and safe to be easily collected [189]. Some allergic diseases were studied by analyzing the nasal secretions with the use of combined methods, including electrochemistry [190,191].

Unfortunately, no electrochemical MSAs/MSSs were suggested for the assessment of seminal, nipple aspirate, interstitial fluids and nasal lavage and bronchoalveolar lavage fluids until present. However, we believe that the MSAs based on electrochemical affinity biosensors made of relevant bioreceptors might be suitable for the protein biomarker analysis in these fluids as well.

### 3.2. Sweat, Body Odour and Wearable Sensors

Basically, sweat is a mixture of several electrolytes, which is produced by the body (eccrine glands and apocrine glands) for a simple purpose: thermoregulation. Nevertheless, it was discovered that composition of sweat contains the medical information important for the diagnosis of numerous diseases. Sweat contains various potential analytes: metabolites (lactate, glucose, urea), amino acids, cortisol, neuropeptide Y, cytokines (IL1α, IL1β, IL6, TNF-α, IL8, and TGF-β), acetone (and VOCs), and pH [109]; they can indicate different skin diseases, diabetes, kidney failure, chronic diseases, aging, and even stress and depressive disorders [109]. There are several approaches to sweat analysis: (i) in vitro, when sweat samples are collected from patient put under conditions causing the extensive sweating (e.g., in dry sauna), and are analyzed by the MSS/MSA in the laboratory [192]; (ii) in situ, when sweat biomarkers are accessed by the MSS/MSA attached to the skin surface and functioning in “on” or “off” modes [193] (Figure 10a). The latter one forms a large group of wearable (bio)electronics, which have been extensively studied and developed within the last years. The attention drawn to the wearable electronics can be illustrated by the fact that some of the recently published reviews, concerned with this topic [134,193,194,195,196,197,198,199], have already reached more than 100 citations within one year.

Moreover, besides the analysis of dissolved biomarkers in the sweat, it is possible to analyze VOCs emitted by the skin, which reflect the metabolic condition of an individual, and can provide valuable information for health monitoring [200]. Regarding the sweat analysis in vitro, we were able to find only one report describing an electrochemical MSA. The potentiometric MSA consisted of solid contact ISEs for Na^+^, K^+^ and Ca^2+^ ions (valinomycin served as a natural ionophore for K^+^), proposed for the analysis of sweat of athletes [192]. The MSA allowed simultaneous measurement of three cations within physiologically relevant ranges; moreover, it has shown good effectiveness in monitoring of the decrease in the amounts of electrolytes down to the danger level that is highly important to control the state of an athlete after exhausting trainings. The lack of the reports on the MSAs or e-tongues for the analysis in vitro can, possibly, be explained by the complexity of proper sampling technique, which makes the analytical platforms for analysis in situ more advantageous in this regard. Although, we believe that certain existing electrochemical MSAs or e-tongues, which were briefly discussed by Kirsanov et al. in [113], might be suitable for the analysis of the ionic composition of sweat. 

Regarding sweat analysis in situ, there are several important requirements for any analytical platform: (i) biocompatibility (no skin irritants or toxic compounds in the architecture (or at least within the platform/skin interface) of the analytical platform); (ii) high stability; (iii) flexibility; and (iv) selectivity for the targets. Apart from physical sensors (to assess temperature, heart rate, movements etc.), electrochemical sensors for sweat analysis consist not only of the electrode arrays and flexible base/support, but also of particular sweat delivery and signal transduction systems for wireless communication with the e-device for data collection and processing (the latter is optional) [201]. The sweat sampling for this kind of analytical platforms is usually carried out using specific absorbing materials or by local chemical stimulation of the sweat glands (the process is known as iontophoresis), because the medical screenings provide better results for ‘equilibrium’ sweat directly produced by the body [201,202]. Furthermore, sweating rate, sweat evaporation and contamination should be taken into account, because they have a significantly impact on the chemical composition of sweat. 

There are numerous studies of wearable biosensors for single analyte detection reported in the literature [193,194,196]. Some of the reported flexible biosensors are integrated in the MSA in combination with chemical sensors. One of such an MSA is a smart wristband with two integrated amperometric biosensors with GOx and lactate oxidase (LOx), applied for the assessment of glucose and lactose, respectively, two solid contact ISEs sensitive to K^+^ and Na^+^ ions, and a resistance-based temperature sensor made of Cr/Au metal microwires [203]. A water-absorbent thin rayon pad was placed between the electrodes and the skin surface; it was used for absorption and maintenance of sufficient amount of sweat (as reported by the authors, about 10 µL of the sweat was enough to provide stable sensor readings). The MSA was able to operate over a period of four weeks, when the MSA was stored between the measurements at 4 °C. It allowed measurements of the metabolites and cations within the physiologically relevant range: [Na^+^] of 20–120 mM; [K^+^] of 2–16 mM; glucose concentration of 0–200 µM; and lactate concentration of 2–30 mM. Another MSA consisted of an amperometric biosensor with GOx, impedimetric humidity sensor, potentiometric pH sensor, resistance-based strain gauge, heater and temperature sensors, was made in the form of a wearable patch [204,205]. The patch was customized for diabetes patients, allowing not only monitoring of physiological changes, but also providing treatment feedback due to additionally included polymeric microneedles that can be thermally activated for transcutaneous drug delivery. A similar MSA was proposed in the form of a wearable smart band. It consisted of an amperometric glucose biosensor and an array of the sensors for continuously monitoring of vital signs (i.e., heart rate, blood oxygen saturation level, and physical activity) [206]. Such a smart band was tested for the tracking of physiological activity of the athletes during intense exercise activities to prevent hypoglycemic shock. A special MSA for the runners was developed to monitor glucose, lactate, uric acid and urea levels in human perspiration and can be potentially used in sports medicine [207]. This MSA consisted of four piezoelectric biosensors based on ZnO interdigitated electrodes with immobilized LOx, GOx, uricase and urease. 

The majority of the reported electrochemical MSAs for sweat analysis and wearable electronics are composed of electrodes based on metals, carbon materials or artificial conductive polymers [198,202,208]. Natural compounds usually serve as the support and base materials in this kind of MSAs. For example, the paper was used for the formation of microfluidic pads, that, combined with a porous adhesive textile, facilitated the access of sweat to the ISEs surfaces for multiplex measurement of the ion content [209]. The hydrogels, based on different cross-polymerized peptides or carbohydrates, ensure better electrical communication between metal electrodes and the skin surface. For example, wearable electrochemical MSAs with the pad of pilocarpine hydrogel, consisted of pilocarpine nitrate and agar gel, were applied for the monitoring of a methylxanthine drug and caffeine [210]. This kind of analysis is important in doping control and precision medicine, because it helps medical doctors to tailor drug dosages, track patients’ compliance to prescriptions, and understand the complex pharmacokinetics of drugs.

A huge area of wearable electronics based on the MSAs is developed for recording of electrocardiograms (ECG), electroencephalograms (EEG), electrooculograms (EOG), and electromyograms (EMG). The MSAs consist of the array of several sensors with an outer energy supply (Figure 10b) [211,212,213]. Some of them are the advanced multifunctional wearable textiles (also known as e-textiles), in some cases made of natural materials, such as cotton or silk. E-textiles include not only the sensor array, but also the energy supply units (for energy harvesting and storing) and communication unit (for signal transduction and communication) (Figure 10c) [214]. 

Wearable e-textiles allow one to monitor not only ECG, EEG, but also pressure, stretch and chemical (bio)sensors enable the detection of body motion, physical activity and/or metabolic changes, which are (in)directly monitored by the sweat and/or body odor analysis [213,215,216]. Cotton and silk are widely used in the e-textile design due to their renewability, degradability, and mechanical durability; they can also adhere to conductive compounds and are suitable for many knitting technologies [217,218,219]. 

Cotton and PANI composites were applied in the sensing of NH_3_ vapors [30], and in the design of wearable sensors for the monitoring of human body motion [220]. It was shown recently that cotton fabric modification with nanostructured ZnO resulted in the enhancement of ultraviolet protection factor, and in the applicability as gas sensors towards such VOCs as acetaldehyde, ammonia, ethanol [221] and NO_2_ [219]. The modification of the cotton fabrics with PEDOT:PSS resulted in the development of a gas sensor toward acetone [222]. Silk, modified with different conductive polymers such as polypyrrole (PPy) [223], PANI [224], polythiophene [225], as well as with metal [226] or metal oxides [227], and carbon materials [228] was used as a (bio)electrochemical sensor for H_2_O_2_, NH_3_, ascorbic acid, glucose, and dopamine. Such kinds of special patterning of functional compounds on the silk can be integrated in the MSA/MSS and potentially applied in e-textile development.

### 3.3. Multisensing for Pharmaceutical Manufacturing

The high costs of a new product (drug) development are explained not only by the expensive (bio)synthesis and purification of the reaction products, but also by the clinical trials, expensive monitoring systems, quality requirements, ethical concerns, and even featuring of taste and flavor properties of drug. Electrochemical MSAs, e-tongues and e-noses [229,230] have become good alternative methods used in the development and production of new pharmaceuticals, potentially replacing gas chromatography, HPLC, NMR, fluorescence, ELISA, human panels and animal models. Potentiometric e-tongues that are partially composed of ISEs with natural ionophores (valinomycin and nonactin [70,231]) and assisted by proteinase K demonstrated promising results in indirect monitoring of protein biosynthesis in bioreactors [232], protein purity control [233] and protein quantity evaluation [234]. These studies demonstrated that one e-tongue could be applied in fast protein quality assessment at different production stages. Instead of human or animal taste panels, an e-tongue of similar composition [235] was applied for the analysis of drug bitterness and dissolution useful for the control of a palatability of the formulations. Other e-tongues containing natural ionophores were applied in the classification of pharmaceutical samples by their active compound and taste masking technique [236], or in the toxicity evaluation of herb preparations [237].

Similarly, bio-e-noses with olfactory receptors and olfactory cells were also applied in the evaluation of drug odor [6]. Very recently, several reviews on electrochemical e-tongues and e-noses in pharmaceutical manufacturing were published [229,230,238,239,240]. MSAs or e-tongues with the bioreceptors, suitable for studying the protein-protein, protein-drug, protein-nucleic acid, protein-lipid interactions, were not reported so far for pharmaceutical applications. Although, the electrochemical analytical devices with single sensor for single analyte detection were suggested in this field [241,242,243]. 

MSAs, based on electrochemical microchips and immobilized cells (‘cell-on-a-chip’), tissues (‘tissue-on-a-chip’), organs (‘organ-on-a-chip’), and even the human body (‘body/human-on-a-chip’) were applied for the screening of the effects of novel pharmaceutical compounds. The most convenient targets for analysis by electrochemical methods in pharmaceutical industry are ion channel proteins [244]. These proteins play essential roles in many cellular functions of human body, such as signal transmission in the nervous systems and mass transportation across the cell membrane [245]. A number of common diseases, such as epilepsy, arrhythmia, are found to be connected to ion channel dysfunctions [246]. The method, known as patch clamp electrophysiology, even became a gold standard in pharmacological investigations, since it allows measuring ion channel activity inside of the whole cell in real time; furthermore, it is flexible for the modifications and provides the best quality of obtained data [247] (Figure 11). This method is based on the measurements of current, generated by ions passing via plasma membrane channels. Such ions’ movements induces fluctuations of the potential, which can be measured with the glass microelectrodes patched onto cellular surfaces at GΩ seal resistance [248]. At present, many electrophysiological platforms are commercially available [247], providing simultaneous monitoring of up to 64 independent reactions; although, the development of a complete platform is under improvement and optimization. Recently, a new design of microchips, which solves the problem of miniaturization of the electrophysiological platforms and simplifies the microfluidic connections, has been suggested [244]. 

Modern data analysis techniques, e.g., artificial neural networks [249], allow the defragmentation of the noise from the raw electrode responses into separate segments, thus, providing the valuable information to a researcher and making possible to assess automatically the activity of a complex single molecule (drug) more accurately and rapidly than the conventional methods. Another example of the MSAs used in the study of single cell (or group of the cells) activity is electrochemical bioimaging. So-called large-scale integration (LSI)-based devices operate on the amperometric or potentiometric signal registration systems; they were successfully applied to study the viability of the human cells, depending on physical or chemical factors [250]. 

The general approach of single cell analysis by means of electrochemical methods has been recently reviewed in [251,252,253,254]. Unfortunately, a single cell cannot often represent the behavior of the entire organism, triggered by the treatment by one or several pharmaceutical compounds. Therefore, the development of more sophisticated systems, modeling the behavior of the whole organism, and avoiding the use of the animal models, is required. This issue might be solved by the development of the new biosensor platforms known as ‘tissue-on-a-chip’, ‘organ-on-a-chip’, and ‘body/human-on-a-chip’ [255]. Such kinds of biosensors were proved to be powerful tools for building the pharmacokinetic and pharmacodynamics models to monitor the complex interactions between several cells/tissues/organs caused by the presence pharmaceutical compounds [256]. Several levels of the assessment of the multi-cell behavior are used: (i) analysis of the adherent cells in two-dimensional (2D) cultures, where the cells are grown until they cover the surface of the (micro)chip; (ii) analysis of the three-dimensional (3D) tissue, which was grown on the (micro)chip; (iii) analysis of the several 3D tissues, which were grown on the chip and connected by special controlled junction or junctions. The role of the electrochemical MSAs is mostly related to the first one—the analysis of 2D cell cultures. The toxicity of new drugs, drug carriers and nanoparticles is often evaluated by electrochemical MSSs, because the dielectric properties of living cells can alter the impedance of an electrode on which they are growing. Currently, some impedimetric MSSs have become commercially available: they allow simultaneous monitoring of up to 384 cell cultures grown on a special chip [257]. 

The development of the tissue-on-a-chip is necessary to understand the interactions between the different tissues, and the response of the whole tissue to the treatment by selected compounds. Such a type of the analytical devices cannot be produced without customized microfluidic platforms which maintain the proliferation, growth and viability of the cells, and deliver the compounds to the target place on the tissue [255]. These sensing devices usually consist of an array of the electrodes, which detect changes in the resistance, pH, ions, oxygen concentration, glucose, temperature and strain of the tissues. One of the most popular tissue-on-a-chip devices is a model of the blood brain barrier, which was formed by a continuous endothelium that regulates the exchange between the blood stream and brain [258]. This physiological barrier has to be taken into account in treatment of neurological diseases, as it prevents the entering of most blood-circulating drugs into brain. Models of neuronal networks [259,260], cardiac tissue [73], melanoma [261] and lung cancer [262] were also built on microelectrode arrays. 

The complexity of these analytical devices increases gradually, going from ‘organ-on-a-chip’, ‘multi-organ-on-a-chip’ to ‘body-on-a-chip’ platforms. These microdevices can mimic tissue–tissue interactions that occur as a result of metabolite transfer from one tissue to another in vitro. As a result, it is possible to obtain a model of the human metabolism, which can predict the conversion of a pro-drug into its effective metabolite as well as its subsequent therapeutic actions and toxic side effects [263]. These platforms use several methods (combining electrochemical and optical ones) for the registration of the changes occurring in the organ models. The most important one is a registration of dynamic changes of cellular health or functions at multiple time points, and a monitoring of the environmental conditions in the whole system. Usually electrochemical methods are applied to monitor temperature, pH and oxygen, because the changes in these three parameters strongly affect cellular functions, e.g., metabolic rates, protein folding, extracellular matrix synthesis, cardiac contraction, immune function [264]. The examples of ‘multi-organ-on-a-chip’ platforms are the models of combined cardiac, muscle, neuronal and liver tissues [74,265] or combined intestine, liver, kidney proximal tubule, blood-brain barrier and skeletal muscle tissues [266], multiorgan tumors [267], which all were designed for the drug toxicity assessment during the period up to 28 days (the minimum timeframe in animal studies to evaluate repeat dose toxicity). 

## 4. Conclusions and Further Perspectives

The application of multisensor system or arrays provides new perspectives for medical diagnosis. It can potentially improve routine laboratory diagnostic procedures, reducing their costs and time of analysis; besides, this approach allows continuous online health monitoring. The MSSs/MSAs are especially attractive in this regard due to their noninvasiveness, simplicity and safety of a sampling procedure, as well as the possibility of simultaneous detection of several analytes. Such systems can be integrated in textile, clothing, patches, pills or into contact lenses. As discussed in this review, naturally-occurring materials serve as an excellent MSS/MSA base for the deposition of electrodes or conducting materials, following the requirements of biodegradability.

Since there is a large variety of body fluids containing information about numerous pathologies (e.g., blood and its serum, saliva, urine and sweat are widely used for the analysis of biomarkers, indicating the presence of a disease), the development of new methods and techniques for extracting this information is of great importance. The MSSs/MSAs can be successfully developed or adapted for a particular task or requirements due to the large variability of their components and materials. For example, such body fluids such as tears, wound exudate, cerebrospinal and gastrointestinal fluid, stools and nasal secretions are rarely used for the assessment of biomarkers; nevertheless, several studies describing their potential applicability for this purpose, have been published recently. The use of MSSs/MSAs in these studies might be highly promising; although, several requirements have to be fulfilled. First of all, the sample volume of the body fluids, mentioned above, is typically limited by several mL or even µL that has to be taken into account when designing the MSS/MSA. In this case, the natural bioreceptors seem to be the only option, providing the highest specificity towards the target compounds. Moreover, the recent development of biotechnological tools for the biosynthesis has significantly reduced the costs of bioreceptor production. Nevertheless, the electrode architecture, sample delivery and signal amplification systems, as well as processing of the sensor readouts, still need to be improved, since the performance, stability and natural specificity of the bioreceptors are largely dependent on the measurement procedure.

We believe that our review demonstrates the high potential of natural and naturally-occurring compounds in the development of the MSSs/MSAs and might inspire other researchers to create new analytical systems, based on such promising materials, for various applications in medical diagnosis, health monitoring or pharmaceutical manufacturing. 

## Figures and Tables

**Figure 1 sensors-20-03551-f001:**
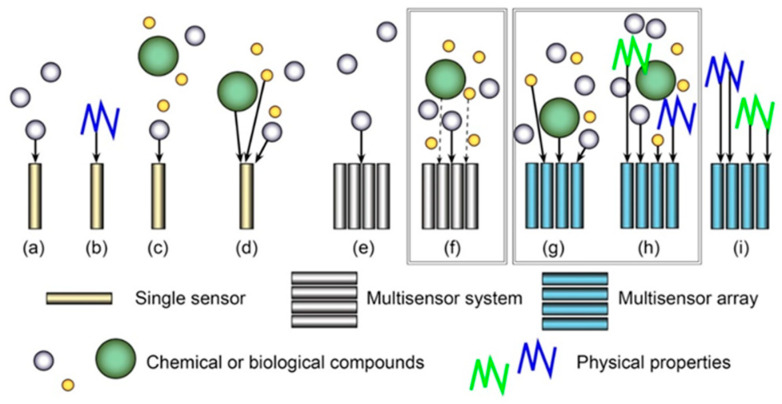
Types of sensors and their ensembles according to the studied characteristics of a sample. Single-sensor (**a**–**d**), multisensor systems (**e**,**f**), multisensor arrays (**g**–**i**) and possible targets for detection. Detailed descriptions are provided in the text.

**Figure 2 sensors-20-03551-f002:**
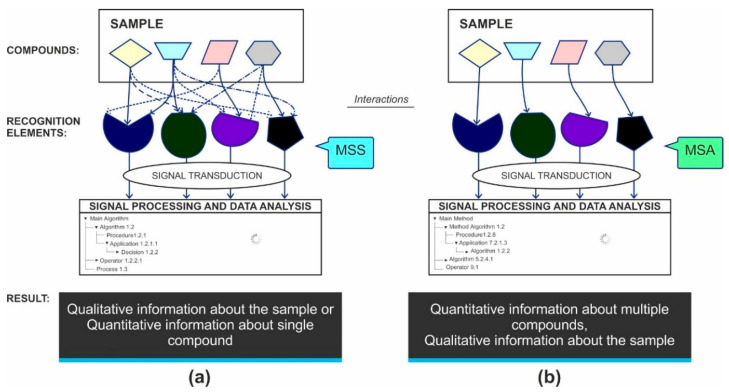
Schematic representation of the sample analysis path for an MSS (**a**) and an MSA (**b**) with recognition elements.

**Figure 3 sensors-20-03551-f003:**
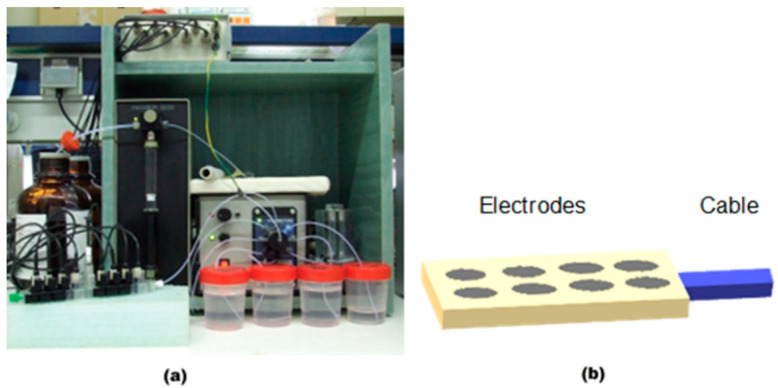
Array of sensors in microfluidic system [13] (**a**); Copyright © 2012 Manel del Valle. Array of microsensors within one electrode (**b**).

**Figure 4 sensors-20-03551-f004:**
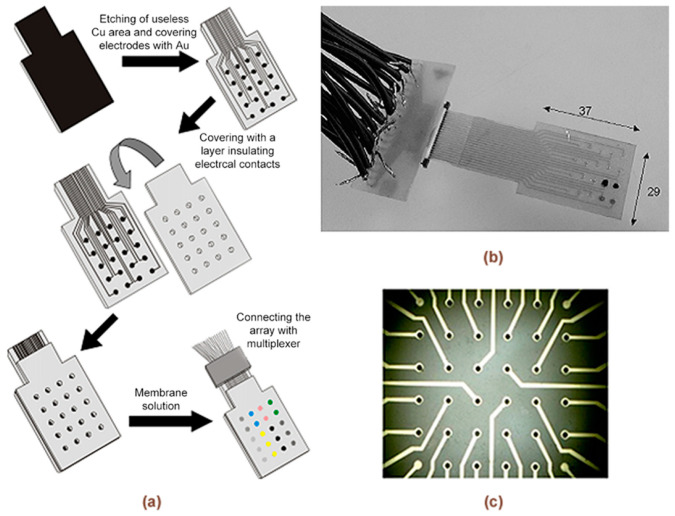
MSA functioning as MSS: preparation by etching and insulation (**a**), photograph of the prepared cable plugged array [23] (**b**), and image of microelectrode array used for cell analysis [24] (**c**). Reproduction permissions granted by Elsevier.

**Figure 5 sensors-20-03551-f005:**
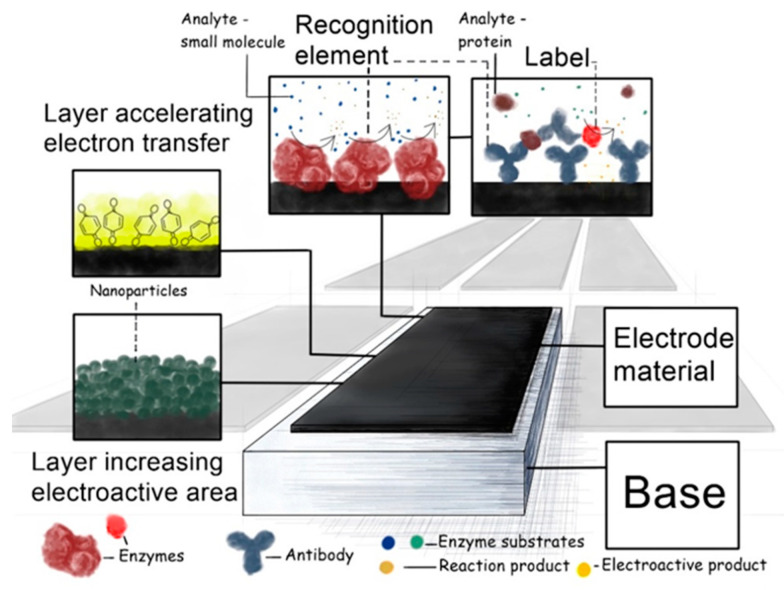
Schematic representation of the possible parts of the MSS/MSA.

**Figure 6 sensors-20-03551-f006:**
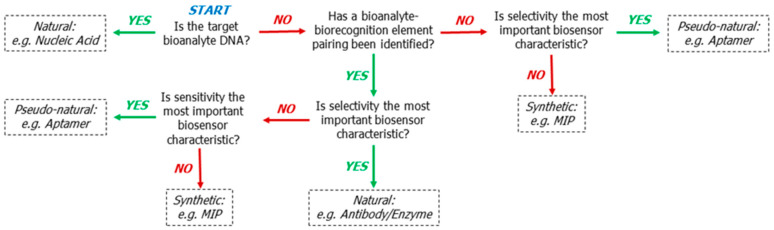
A flowchart for choosing a recognition element for biosensors, adopted from [77]. Reproduction permission granted by ACS Publishing.

**Figure 7 sensors-20-03551-f007:**
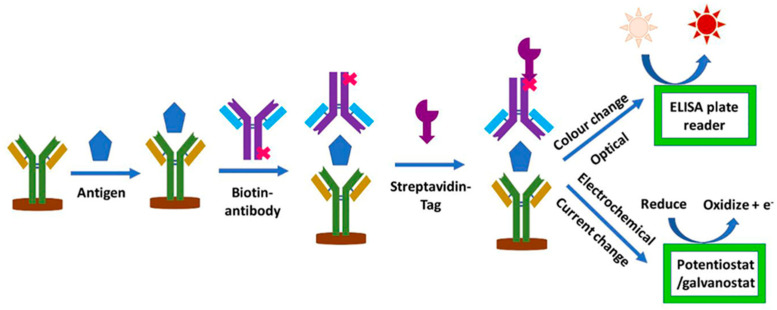
General scheme of the analysis by ELISA. Reused from [83]. Copyright 2018, MDPI AG.

**Figure 8 sensors-20-03551-f008:**
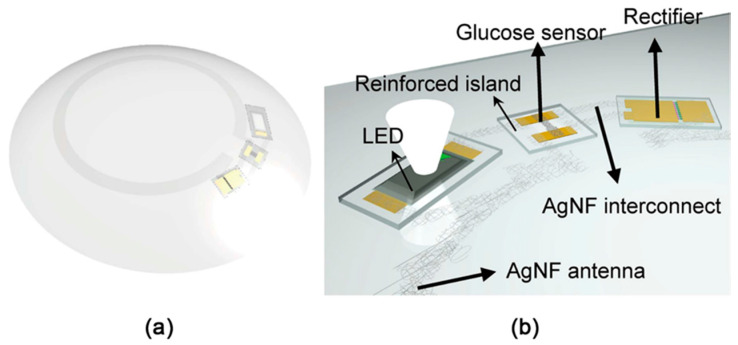
Schematic illustration of the smart contact lens. The contact lens consists of the soft base—hybrid substrate, which is made of photocurable optical polymer with micropatterned Cu layer and silicone elastomere (elastofilcon A). The functional devices integrated in the base are: a rectifier, LED, a glucose sensor and a transparent, stretchable conductor for antenna and interconnects, made of Ag nanofibers (AgNF). Adopted with permission from [135]. Copyright 2018, American Association for the Advancement of Science.

**Figure 9 sensors-20-03551-f009:**
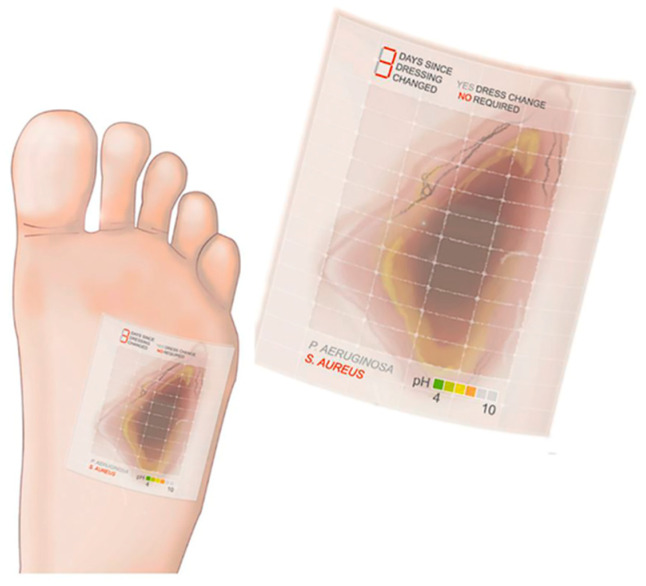
The possible evolution of MSA/MSS, designed for wounds in the future: the MSA/MSS is incorporated into the wound bandage, monitoring the size of the wound, pH changes, bacterial contamination, etc. Reused form [181]. Reproduction permission granted by Elsevier.

**Figure 10 sensors-20-03551-f010:**
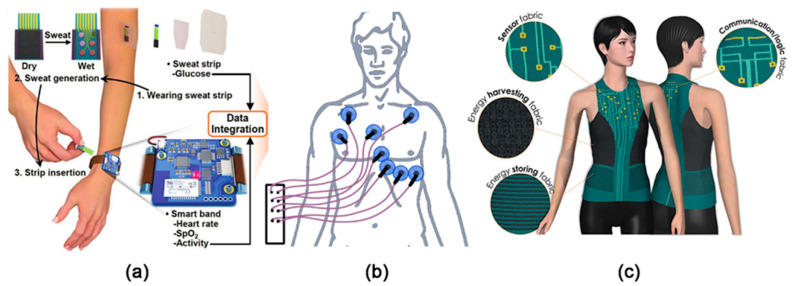
Examples of wearable electronics. Band, patch [206] (**a**), sample of electrode array for ECG, EEG, EOG and EMG (**b**), and for e-textile [214] (**c**). Reproduction permissions were obtained from John Wiley and Sons.

**Figure 11 sensors-20-03551-f011:**
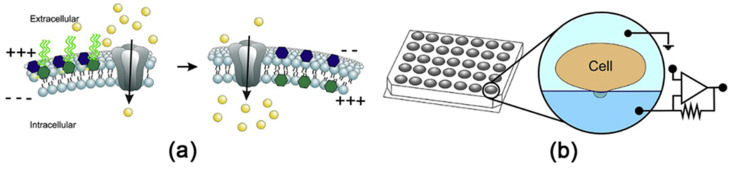
Schematic representation of the patch clamp technique. The bioprocess, which occurs on the interface of the cell membrane due to the action of the pore forming protein and induced the voltage changes (**a**). The example of planar patch clamp high-throughput automated electrophysiology platform (**b**) [247]. Reproduction permission granted by Elsevier.

**Table 1 sensors-20-03551-t001:** Summary of MSS/MSA application for blood analysis.

Measured Parameters	Method	MSS/MSA	Naturally-Occurring Compound	Ref.
CA19-9 and CA125, H_2_O_2_	Amperometry	Cellulose acetate membrane to co-immobilize thionine as a mediator and two kinds of antigens on two carbon electrodes of a screen-printed chip.	CA19-9 and CA125 antigens	[34]
α-fetoprotein, CEA	ELISA	Poly(o-phenylenediamine)/Au nanocomposite and poly(vinyl ferrocene-2-aminothiophenol) (poly(vinyl ferrocene-ATP))/Au nanocomposite	Antibodies for α-fetoprotein, CEA	[87]
CEA, mucin-1	DPV	Au ion electrochemical aptamer labels and Ru(NH_3_)_6_^3+^ electronic wires	DNA, aptamers	[88]
CEA, neuron-specific enolase	Amperometry	SPEs on paper modified with wither amino functional graphene-Thionin-AuNPs or Prussian blue-PEDOT-AuNPs nanocomposites	DNA, aptamers	[89]
MMP2, MMP7	EIS	Au/glass WE and PDMS microfluidic system	Peptides and amino acids for peptide synthesis	[93]
Dobutamine, amlodipine, paracetamol, ascorbic acid	SWV	Layered composite based on layer-by-layer modification of a glassy carbon electrode with multi-walled carbon nanotubes, ionic liquid crystal, graphene and 18-Crown-6: GC/CNT/ILC/RGO/CW	-	[96]
Adenine, guanine	DPV	MnO_2_ nanosheets and ionic liquid functionalized graphene bound to a polydopamine membrane on glassy carbon: PDA/MnO_2_/IL-GR/GCE	MnO_2_	[98]
Glucose level, temperature, blood pressure	Galvanic skin response with photoplethysmogram signal measurement	Non-invasive galvanic sensor	Enzyme glucose oxidase	[103]
Heart work, glucose level, motion	Wearable wireless sensing	Electrocardiogram, three-axis accelerometer, continuous blood glucose monitor	Enzyme glucose oxidase	[107]
IL-β1, TNF-α cytokines	Steady-state current measurements	SPCE immunosensors modified with 4-carboxyphenyl-functionalized double-walled CNTs	Antibodies to IL-β1, TNF-α	[108]

**Table 2 sensors-20-03551-t002:** Summary of MSS/MSA application for analysis of urine.

Measured Targets	Method	MSA	Naturally Occurring Compounds	Ref.
Biomarkers NUMA1, CFHR1	ELISA	Two off-site matrices composed of 3D polymethyl methacrylate (PMMA) sheets and 2D polycarbonate (PC) membranes modified with the required antibodies.	Antibodies to NUMA1, CFHR1	[110]
Na^+^, K^+^, NH_4_^+^, Ca^2+^, Mg^2+^, Cl^−^, SO_4_^2-^, PO_4_^3-^, urate, creatinine	Potentiometry	A set of ion-selective sensors	Enzyme ureate oxidase	[114]
Various cations and anions	Potentiometry; “fingerprint” detection	A set of ion-selective sensors	-	[115]
Na^+^, K^+^, NH_4_^+^, H^+^, urea	Potetciometry	An array of ISEs + biosensor	Enzyme ureate oxidase	[117]
Urea, creatinine	Potentiometry	Biosensor array	Enzyme ureate oxidase	[118]
Glucose, creatinine, uric acid	DPV	A disposable electrochemical paper-based multiplexed working electrode microfluidic device modified with ferrocenecarboxylic acid and glucose oxidase, non-complexed Fe^3+^, and carbon black NPs for glucose, creatinine, and uric acid, respectively.	Enzyme glucose oxidase	[122]
Morphine, benzoylecgonine, tetrahydrocannabinol	SWV	Immunosensor array	Bovine serum albumin, antibodies: anti-morphine monoclonal antibody, anti-tetrahydrocannabinol monoclonal antibody, anti-benzoylecgonine monoclonal antibody	[123]

**Table 3 sensors-20-03551-t003:** Summary of MSS/MSA application for analysis of saliva.

Measured Targets	Method	MSA	Naturally Occurring Compounds	Ref.
IL-8 mRNA, IL-8 protein	Amperometry	Array of 16 Au electrode chips modified with polypyrrole, streptavidin-modified dendrimer NPs	Streptavidin, biotin, human interleukin-8 monoclonal antibody	[126]
IL-8 mRNA, IL-8 protein	Amperometry	SPCE immunosensors	human interleukin-8 monoclonal antibody	[127]
IL-β1, TNF-α cytokines	Steady-state current measurements	SPCE immunosensors modified with 4-carboxy-phenyl-functionalized double-walled CNTs	Mouse TNFα capture antibody, biotinylated goat TNFα detection antibody, mouse IL1β capture antibody, biotinylated goat ILβ1 detection antibody	[108]
hGH and PRL	Amperometry	CNT/SPCEs modified with poly(ethylene-dioxy-thiophene) (PEDOT), AuNPs and corresponding hGH and PRL antibodies	Mouse monoclonal anti-hGH antibody, mouse monoclonal anti-PRL antibody	[129]
GHRL and PYY	DPV	Dual SPCEs modified with reduced graphene oxide, diazonium salt of 4-aminobenzoic acid (4-ABA), covalently immobilized antibodies	Ghrelin rabbit polyclonal antibody, biotinylated-GHRL Peptide YY (3–36) purified IgG antibody, peptide YY (3–36), biotinylated-PYY (3–36)	[130]
Na^+^, K^+^, Ca^2+^, Mg^2+^, Cl^−^	Potentiometry	Ion selective electrodes modified with PEDOT	valynomicin	[131]
Na^+^, K^+^, Ca^2+^, Mg^2+^, Cl^−^, SCN^−^, pH	Potentiometry	Au electrode modified electrosynthesized PEDOT:PSS and ion-selective membrane	-	[132]

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
