# Peer review of "Multisensor Systems and Arrays for Medical Applications Employing Naturally-Occurring Compounds and Materials"

_sensors, 2020, doi:10.3390/s20123551_

Round 1

Reviewer 1 Report

The manuscript reports an extensive review about bases and applications of multisensor systems and arrays for the simultaneous analysis of several analytes of medical interest in diverse biological fluids. The information reported is interesting and useful, so its publication is recommended, although a minor revision is required.

Comments and suggestions

  1. Discussion of the importance of natural compounds used in the development of the analytical platforms is poorly integrated in the manuscript. In this sense, It is recommended an additional section which includes a table that concentrate the materials reported, uses and its main features.
  2. Lines 107-109. It seems that second idea of the sentence is out of context.
  3. Line 128. Please adjust the figure position, since it hides part of the figure caption.
  4. Line 162. Improve the quality of Figure 4.
  5. Line 234-235. Please correct sentence drafting
  6. Line 253. Please replace “carbo” with “carbon”
  7. Line 265. Please replace “nanomatelrials” with “nanomaterials”
  8. Check all acronyms were defined.

Author Response

Comment: Discussion of the importance of natural compounds used in the development of the analytical platforms is poorly integrated in the manuscript. In this sense, It is recommended an additional section which includes a table that concentrate the materials reported, uses and its main features.

Authors: We thank the Reviewer for the suggestion. Current manuscript contains a section 2.3: “Materials used in electrochemical MSAs and MSSs”, which is devoted to the general description of typical materials and to the discussion of the role of natural compounds as a part of the multisensor systems and arrays. Aiming to highlight the roles of natural materials in the MSSs/MSAs, some additional information is added to the section 2.3. Moreover, we have rewritten the introduction section and conclusions, where we narrowed down the scope of the manuscript and highlighted the importance of natural compounds and materials in the development of multisensor systems and arrays. However we would like to disagree with the reviewer #1 regarding additional table. In our opinion, one more table would be excessive. We also would like to pay attention of the reviewer, that all the text of the manuscript was corrected by the native English speaker.

Comment: Lines 107-109. It seems that second idea of the sentence is out of context.

Authors: The Reviewer is right. Application to Parkinson’s disease is not related to multisensors. We have eliminated that part of the sentence. 

Comment: Line 128. Please adjust the figure position, since it hides part of the figure caption.

Authors: This has been corrected. 

Comment: Line 162. Improve the quality of Figure 4.

Authors: This is a hard task because we took Figure parts from pdf versions of the articles since other options are not available. But we tried to do our best to adopt pictures or modify them with appropriate software. 

Comment: Line 234-235. Please correct sentence drafting.

Authors: The word “famous” has been replaces with “known” in the indicated sentence. 

Comment: Line 253. Please replace “carbo” with “carbon”. Line 265. Please replace “nanomatelrials” with “nanomaterials”.

Authors: We apologize for the mistyping. These mistakes have been corrected. 

Comment: Check all acronyms were defined.

Authors: These has been done and corrected where definition of acronyms was missing.

Reviewer 2 Report

In their manuscript, the authors have reviewed the current state of microsensor arrays and microsensor systems base on natural and naturally abundant compounds. The review is extensive with almost 300 references but a considerably large number of these citations are actually other, already published reviews. The authors do mention about including also reviews in their manuscript but I don't think this large number of them is acceptable, especially since in many cases they could have referenced the original work instead. 

When it comes to the scope of the review, it should be clarified and focused more for example to include only electrochemical devices (or whatever the authors see fit). There are several reviews covering wearable electronics and I don't think that part in particular is necessary here.

What I find very confusing is that it is not explained clearly in the manuscript what are the "natural and naturally abundant compounds" that the authors talk about. If I was playing the devil's advocate, I could say that for example carbon nanotubes or even single layers of graphene used in sensors are not natural and definitely not naturally abundant - and especially neither of them is a compound. In addition, even though paper is made of naturally occurring materials and enzymes exists in nature as well, it doesn't consequently mean that sensors made by using them would be more biocompatible than something made of synthetic polymers. Moreover, the levels of biocompatibility required from invasive and non-invasive devices are very different. I find that using only the biocompatibility or something being made of carbon as an argument for the scope, especially without proper references, is not enough in this case. 

Finally, the authors should do a thorough language check on the manuscript, including the abstract. The text is written in very informal, non-scientific manner.

As the authors state, multisensor arrays show promise in regard of precision medicine. Finding materials that are environmentally friendly is important from sustainability point-of-view. Thus, a review summarizing the latest related research is interesting and would surely find be relevant to the readers of the Sensors journal. However, in it's current form the manuscript does not give a focused and precise picture of the state-of-the-art in such devices. Thus, my recommendation is that the authors revise the manuscript thoroughly and send it for re-evaluation (major revisions). 

Author Response

Comment: In their manuscript, the authors have reviewed the current state of microsensor arrays and microsensor systems base on natural and naturally abundant compounds. The review is extensive with almost 300 references but a considerably large number of these citations are actually other, already published reviews. The authors do mention about including also reviews in their manuscript but I don't think this large number of them is acceptable, especially since in many cases they could have referenced the original work instead.

Authors: We are very thankful to the Reviewer for raising this question. We partially agree with the Reviewer. Our all cited reviews contain valuable and interesting information related to the particular case mentioned in the manuscript. Thereby each reader could easily found the origin of the scope, which could be under his/her interest. We also would not like to offend other authors of the review works at similar fields. Especially, when we have had experience that other Reviewers commented that we ignored already existing reviews in similar field in our previous review works. Multisensing is very attractive topic, and it has a lot of review articles, thus we included the latest and interesting surveys. Possibly, the number of the cited review papers is higher in current manuscript comparing to the Reviewer #2 expectations. Thereby we have carefully checked the cited review papers, and we have changed them with original research papers in the possible cases. Moreover, we also carefully checked the citations regarding original works, aiming to omit the citation of the review instead. Since we revised all the manuscript carefully, and have rewritten the introduction section and the conclusions, the numeration of the references was changed starting from the reference [7]. The list of added references:

8          Parastar, H.; Kirsanov, D., Analytical figures of merit for multisensor arrays. ACS Sensors 2020, 5, 580-587, doi:10.1021/acssensors.9b02531.

11        Ratas, I.; Pyragas, K., Macroscopic self-oscillations and aging transition in a network of synaptically coupled quadratic integrate-and-fire neurons. Physical Review E 2016, 94, doi:10.1103/physreve.94.032215.

20        Bard, A. J.; Faulkner, L. R., Basic potential step methods. In Electrochemical methods. Fundamentals and applications, 2nd ed.; John Wiley & Sons: 2001; pp 156-225.

22        Bard, A. J.; Faulkner, L. R., Potential sweep methods. In Electrochemical methods. Fundamentals and applications, 2nd ed.; John Wiley & Sons: 2001; pp 226-260.

23        Bard, A. J.; Faulkner, L. R., Techniques based on concepts of impedance. In Electrochemical methods. Fundamentals and applications, 2nd ed.; John Wiley & Sons: 2001; pp 368-416.

24        The Free Dictionary [Internet]. "Natural compound". Webster's Revised Unabridged Dictionary, G. & C. Merriam Co., 1913 [cited 7 Jun. 2020]. Available from: https://www.thefreedictionary.com/Natural+compound.

48        Al-Oqla, F. M.; Sapuan, S. M.; Anwer, T.; Jawaid, M.; Hoque, M. E., Natural fiber reinforced conductive polymer composites as functional materials: A review. Synth. Met. 2015, 206, 42-54, doi:10.1016/j.synthmet.2015.04.014.

81        Caduff, A.; Hirt, E.; Feldman, Y.; Ali, Z.; Heinemann, L., First human experiments with a novel non-invasive, non-optical continuous glucose monitoring system. Biosens. Bioelectron. 2003, 19, 209-217, doi:10.1016/s0956-5663(03)00196-9.

The list of deleted references with numeration at the unrevised version of the manuscript:

7          Winquist, F., Voltammetric electronic tongues – basic principles and applications. Microchim. Acta 2008, 163, 3-10, doi:10.1007/s00604-007-0929-2.

8          Mohamed, E. I.; Abdel-Mageed, S. M., The electronic tongue – basic principles and medical applications. J. Biophys. Biomed. Sci. 2010, 3, 290-295.

10        Wang, J.; Wu, C.; Hu, N.; Zhou, J.; Du, L.; Wang, P., Microfabricated electrochemical cell-based biosensors for analysis of living cells in vitro. Biosensors 2012, 2, 127-170, doi:10.3390/bios2020127.

11        Latha, R.; Lakshmi, P., Electronic tongue: An analytical gustatory tool. J. Adv. Pharm. Technol. Res. 2012, 3, 3-8, doi:10.4103/2231-4040.93556.

12        Tahara, Y.; Toko, K., Electronic tongues – A review. Sensors J. IEEE 2013, 13, 3001-3011, doi:10.1109/JSEN.2013.2263125.

14        Physiology of taste (Available online: www.vivo.colostate.edu/hbooks/pathphys/digestion/pregastric/index.html). In Pathophysiology of the digestive system Bowen, R., Ed. Colorado state university: Colorado, US, 2020.

18        Pyragas, K., A twenty-year review of time-delay feedback control and recent developments. In Proceedings of international symposium on nonlinear theory and its applications (NOLTA2012), Spain, 2012; pp 683-686.

26        Grieshaber, D.; MacKenzie, R.; Vörös, J.; Reimhult, E., Electrochemical biosensors - Sensor principles and architectures. Sensors (Basel) 2008, 8, 1400-1458.

27        Lisdat, F.; Schäfer, D., The use of electrochemical impedance spectroscopy for biosensing. Anal. Bioanal. Chem. 2008, 391, 1555-1567, doi:10.1007/s00216-008-1970-7.

28        Wang, Y.; Ye, Z.; Ying, Y., New trends in impedimetric biosensors for the detection of foodborne pathogenic bacteria. Sensors (Basel) 2012, 12, 3449-3471, doi:10.3390/s120303449.

122      Shen, L. L.; Zhang, G. R.; Etzold, B. J. M., Paper‐based microfluidics for electrochemical applications. ChemElectroChem 2019, 7, 10-30, doi:10.1002/celc.201901495.

195      Munje, R. D.; Muthukumar, S.; Panneer Selvam, A.; Prasad, S., Flexible nanoporous tunable electrical double layer biosensors for sweat diagnostics. Sci. Rep. 2015, 5, 14586, doi:10.1038/srep14586.

219      Yan, L.; Bae, J.; Lee, S.; Roh, T.; Song, K.; Yoo, H.-J., A 3.9 mW 25-electrode reconfigured sensor for wearable cardiac monitoring system. IEEE J. Solid-State Circuits 2011, 46, 353-364, doi:10.1109/jssc.2010.2074350.

220      Hoffmann, K.; Ruff, R., Flexible dry surface-electrodes for ECG long-term monitoring. In 29th Annual International Conference of the IEEE Engineering in Medicine and Biology Society, 2007; pp 5739-5742, doi:10.1109/IEMBS.2007.4353650.

221      Spinelli, E.; Haberman, M., Insulating electrodes: a review on biopotential front ends for dielectric skin–electrode interfaces. Physiol. Meas. 2010, 31, S183-S198, doi:10.1088/0967-3334/31/10/s03.

222      Logothetis, I.; Fernandez-Garcia, R.; Troynikov, O.; Dabnichki, P.; Pirogova, E.; Gil, I., Embroidered electrodes for bioelectrical impedance analysis: impact of surface area and stitch parameters. Meas. Sci. Technol. 2019, 30, 115103, doi:10.1088/1361-6501/ab3201.

223      Tsukada, S.; Nakashima, H.; Torimitsu, K., Conductive polymer combined silk fiber bundle for bioelectrical signal recording. PLoS One 2012, 7, e33689, doi:10.1371/journal.pone.0033689.

232      Zhang, L.; Li, H.; Lai, X.; Gao, T.; Liao, X.; Chen, W.; Zeng, X., Carbonized cotton fabric-based multilayer piezoresistive pressure sensors. Cellulose 2019, 26, 5001-5014, doi:10.1007/s10570-019-02432-x.

234      Zhang, M.; Wang, C.; Wang, H.; Jian, M.; Hao, X.; Zhang, Y., Carbonized cotton fabric for high-performance wearable strain sensors. Adv. Funct. Mater. 2017, 27, doi:10.1002/adfm.201604795.

235      Souri, H.; Bhattacharyya, D., Highly sensitive, stretchable and wearable strain sensors using fragmented conductive cotton fabric. J. Mater. Chem. C 2018, 6, 10524-10531, doi:10.1039/c8tc03702g.

236      Chen, X.; An, J.; Cai, G.; Zhang, J.; Chen, W.; Dong, X.; Zhu, L.; Tang, B.; Wang, J.; Wang, X., Environmentally friendly flexible strain sensor from waste cotton fabrics and natural rubber latex. Polym. 2019, 11, doi:10.3390/polym11030404.

237      Ma, Z.; Xu, R.; Wang, W.; Yu, D., A wearable, anti-bacterial strain sensor prepared by silver plated cotton/spandex blended fabric for human motion monitoring. Colloids Surf. Physicochem. Eng. Aspects 2019, 582, doi:10.1016/j.colsurfa.2019.123918.

238      Asfar, Z.; Nauman, S.; Ur Rehman, G.; Mumtaz Malik, F.; Ayaz, Y.; Muhammad, N., Development of flexible cotton-polystyrene sensor for application as strain gauge. IEEE Sens. J. 2016, 16, 8944-8952, doi:10.1109/JSEN.2016.2618726.

239      Qu, J.; He, N.; Patil, S. V.; Wang, Y.; Banerjee, D.; Gao, W., Screen printing of graphene oxide patterns onto viscose nonwovens with tunable penetration depth and electrical conductivity. ACS Appl. Mater. Interfaces 2019, 11, 14944-14951, doi:10.1021/acsami.9b00715.

240      Wang, X.; Gu, Y.; Xiong, Z.; Cui, Z.; Zhang, T., silk-molded flexible, ultrasensitive, and highly stable electronic skin for monitoring human physiological signals. Adv. Mater. 2014, 26, 1336-1342, doi:10.1002/adma.201304248.

Comment: When it comes to the scope of the review, it should be clarified and focused more for example to include only electrochemical devices (or whatever the authors see fit). There are several reviews covering wearable electronics and I don't think that part in particular is necessary here.

Authors: We are grateful for the comment. Aiming to clarify the scope of the review, we have rewritten introduction section. Some multisensor arrays, mentioned in this manuscript, belong to the wearable electronics. However, the combination of electrochemical sensors and electronic sensors are often used for medical analysis, e.g. for the tracking of the sweat and body odour composition together with the heart bit and movement. Thereby we would like to include these MSAs in the review. Following recommendations, we deleted discussions, which were related only to the electronics without electrochemical sensors in the section 3.2.  

Comment: What I find very confusing is that it is not explained clearly in the manuscript what are the "natural and naturally abundant compounds" that the authors talk about. If I was playing the devil's advocate, I could say that for example carbon nanotubes or even single layers of graphene used in sensors are not natural and definitely not naturally abundant - and especially neither of them is a compound.

Authors: We agree with the Reviewer. We didn't give a very accurate and clear description of the term “naturally abundant compounds”. “Natural” material or compound mentioned in this review is a compound/material, which can be produced by living organism. Thereby the definition “naturally abundant” was changed to the “naturally-occurring” in the whole text of the manuscript, including a title. Additional clarifications were added to the section 2.3.

Comment: In addition, even though paper is made of naturally occurring materials and enzymes exists in nature as well, it doesn't consequently mean that sensors made by using them would be more biocompatible than something made of synthetic polymers. Moreover, the levels of biocompatibility required from invasive and non-invasive devices are very different. I find that using only the biocompatibility or something being made of carbon as an argument for the scope, especially without proper references, is not enough in this case.

Authors: The Reviewer is right. As mentioned above, we have rewritten the introduction and conclusion sections and have added some other pros of the naturally occurring materials and compounds at certain discussions.  

Comment: Finally, the authors should do a thorough language check on the manuscript, including the abstract. The text is written in very informal, non-scientific manner.

Authors: The language has been corrected in the revised manuscript by native speaker.

Reviewer 3 Report

Although the subject matter is appropriate for the Journal and the approach is of potential interest to the readers of the journal, but but it requires revision.

  1. It would be better to address different types of Multisensor Systems and Arrays as one review article.
  2. Despite the large number of references, some of them have not been mentioned or discussed in this review such as:

https://pubs.acs.org/doi/pdf/10.1021/acssensors.9b02531

Author Response

Comment: It would be better to address different types of Multisensor Systems and Arrays as one review article.

Authors: We are very thankful to the Reviewer for the comment. We are not quite sure that we got the question right. The scope of this review manuscript was to overview multisensor systems and arrays constructed using naturally-occurring compounds and materials. 

Comment: Despite the large number of references, some of them have not been mentioned or discussed in this review such as:https://pubs.acs.org/doi/pdf/10.1021/acssensors.9b02531

Authors: We apologize for missing the important article. The paper has been cited in the revised manuscript as reference [8].

Round 2

Reviewer 2 Report

The authors have revised their manuscript and improved it significantly. I appreciate especially the work they have done to include original sources as references and the language editing. 

There are still a couple of minor comments that I hope the authors would address before the publication of the manuscript:

Introduction

- Electrochemical sensors are mentioned only at the very end of the introduction. For readers familiar with electrochemistry it is probably obvious why they are such powerful tools for sensors but to serve the rest of the audience, it would be good if there was some kind of introduction or justification why the focus is on them.

Row 202:

"In order to depose or to integrate an electrode in an MSS or MSA, it has to be made of an insulating material."
- This sentence gives the impression that the electrode needs to be made on an insulating material even though I think it is the MSS/MSA base that you mean here.
- Next sentence: use "rigid" (=not flexible) instead of "solid" (opposite of solid is either gaseous or liquid and I think such sensors might not be very popular.)

Section 2.3.6

- It would be better to change the title and focus if the section here to signal amplification in general. Typically, enzymes are not called labels but they can be labeled themselves to enhance the signal. In particular, in reference 78, enzymes are not called labels. "The enzymes of this kind can produce a high concentration of electroactive compounds in a short time period; then, these compounds undergo oxidation or reduction within the electrode material." This is actually just the conventional description for an enzymatic electrochemical biosensor and has nothing to do with labeling.

Chapter 3:

- I feel that in all of the sections the focus is mostly on what was detected. Please highlight the role of the natural materials here as well as that is the main scope of the manuscript. It is not clear from the tables in this chapter that the sensors presented in them are based on natural materials.

Figures

- Add descriptions of a-i to Fig 1.

- Parts of texts are still very small in some of the figures (especially Fig 2, 10, 11). If possible, please increase the size.

Author Response

Comment: The authors have revised their manuscript and improved it significantly. I appreciate especially the work they have done to include original sources as references and the language editing. There are still a couple of minor comments that I hope the authors would address before the publication of the manuscript.

Authors: We are very thankful for the comment. We will try our best to correct those minors issues. 

Comment: Introduction - Electrochemical sensors are mentioned only at the very end of the introduction. For readers familiar with electrochemistry it is probably obvious why they are such powerful tools for sensors but to serve the rest of the audience, it would be good if there was some kind of introduction or justification why the focus is on them.

Authors: We appreciate the Reviewer’s observation. This is our distraction, we apologize for it. The additional information has been added to Introduction of the revised manuscript. 

Comment: Row 202: "In order to depose or to integrate an electrode in an MSS or MSA, it has to be made of an insulating material."- This sentence gives the impression that the electrode needs to be made on an insulating material even though I think it is the MSS/MSA base that you mean here.

Authors: The Reviewer is right. The sentence is the first sentence of the Section 2.3.1 “MSS/MSA base”. The misleading sentence has been rewritten to “Coupling of the MSSs/MSAs with the electrochemical transducers provides many advantages comparing to other analytical methods, e.g., capability to analyze unmodified samples, portability, and possibility of their miniaturization” 

Comment: - Next sentence: use "rigid" (=not flexible) instead of "solid" (opposite of solid is either gaseous or liquid and I think such sensors might not be very popular.)

Authors: We are thankful to the Reviewer for the valuable suggestion. This has been changed in the text of the revised manuscript. 

Comment: Section 2.3.6- It would be better to change the title and focus if the section here to signal amplification in general. Typically, enzymes are not called labels but they can be labeled themselves to enhance the signal. In particular, in reference 78, enzymes are not called labels.

Authors: We appreciate the Reviewer for raising this discussion. We partially agree with the Reviewer; actually, the term “Label” and its definition are dependent on the field of the expertize. For example, in the case of ELISA, more often the definition “enzyme conjugate” is used. However, the terms as “enzymatic tag”, “enzyme tag”, “enzyme label” are found either in the literature or in the web-pages of the suppliers. Moreover, the term “Enzyme label” is known already more than 30 years, especially in the field of electrochemical detection, e.g. the review by G. Sitta Sittampalam and George S. Wilson Tr. Anal. Chem. 3, 1984, 96 (doi.org/10.1016/0165-9936(84)87086-7). Number of recent articles are also using term “enzyme label”, e.g. Analyst 2019, 144, 4589; RCS Adv. 2019, 9, 23658, etc. Aiming to avoid misunderstandings, a clarification is added to the revised manuscript in the section 2.3.6.  

Comment: "The enzymes of this kind can produce a high concentration of electroactive compounds in a short time period; then, these compounds undergo oxidation or reduction within the electrode material." This is actually just the conventional description for an enzymatic electrochemical biosensor and has nothing to do with labeling.

Authors: The action of the enzyme label is the same as that of the electrochemical biosensor of the first generation, therefore, the description is the same and it has been left in the revised manuscript unchanged. 

Comment: Chapter 3:- I feel that in all of the sections the focus is mostly on what was detected. Please highlight the role of the natural materials here as well as that is the main scope of the manuscript. It is not clear from the tables in this chapter that the sensors presented in them are based on natural materials.

Authors: We are thankful for the Reviewer’s comment. We were thinking how to add natural or naturally-occurring compounds without overcrowding with information. The best decision in our opinion was to add additional column to all Tables with naturally occurring compounds used for MSS/MSA development. This has been done in Tables 1-3 in the revised version of the manuscript. 

Comment: Figures- Add descriptions of a-i to Fig 1.

Authors: The descriptions have been added to the revised manuscript. 

Comment: - Parts of texts are still very small in some of the figures (especially Fig 2, 10, 11). If possible, please increase the size.

Authors: The text size on Figure 2 is made following recommendation of the artwork of the journal. Moreover, Figure 2 is made only for the illustration of the principal differences of the MSSs and MSAs, therefore, the small-sized text does not include any essential information. Regarding Figures 10 and 11 – the illustrations were taken from the original works, whereby the text size cannot be magnified. Therefore, no additional changes related to this comment have been made in the manuscript.